# GEOMETRIC AND PHYSICAL QUANTITIES IMPROVE E(3) EQUIVARIANT MESSAGE PASSING

**Johannes Brandstetter**[*]
University of Amsterdam
Johannes Kepler University Linz
`brandstetter@ml.jku.at`

**Rob Hesselink**[*]
University of Amsterdam
`r.d.hesselink@uva.nl`

**Elise van der Pol**
UvA-Bosch DeltaLab
University of Amsterdam
`e.e.vanderpol@uva.nl`

**Erik J Bekkers**
University of Amsterdam
`e.j.bekkers@uva.nl`

**Max Welling**
UvA-Bosch DeltaLab
University of Amsterdam
`m.welling@uva.nl`

## ABSTRACT

Including covariant information, such as position, force, velocity or spin is important in many tasks in computational physics and chemistry. We introduce Steerable E(3) Equivariant Graph Neural Networks (SEGNNs) that generalise equivariant graph networks, such that node and edge attributes are not restricted to invariant scalars, but can contain covariant information, such as vectors or tensors. This model, composed of steerable MLPs, is able to incorporate geometric and physical information in both the message and update functions. Through the definition of steerable node attributes, the MLPs provide a new class of activation functions for general use with steerable feature fields. We discuss ours and related work through the lens of *equivariant non-linear convolutions*, which further allows us to pin-point the successful components of SEGNNs: *non-linear* message aggregation improves upon classic *linear* (steerable) point convolutions; *steerable messages* improve upon recent equivariant graph networks that send invariant messages. We demonstrate the effectiveness of our method on several tasks in computational physics and chemistry and provide extensive ablation studies.

## 1 INTRODUCTION

The success of Convolutional Neural Networks (CNNs) (LeCun et al., 1998; 2015; Schmidhuber, 2015; Krizhevsky et al., 2012) is a key factor for the rise of deep learning, attributed to their capability of exploiting translation symmetries, hereby introducing a strong inductive bias. Recent work has shown that designing CNNs to exploit additional symmetries via group convolutions has even further increased their performance (Cohen & Welling, 2016; 2017; Worrall et al., 2017; Cohen et al., 2018; Kondor & Trivedi, 2018; Weiler et al., 2018; Bekkers et al., 2018; Bekkers, 2019; Weiler & Cesa, 2019). Graph neural networks (GNNs) and CNNs are closely related to each other via their aggregation of local information. More precisely, CNNs can be formulated as message passing layers (Gilmer et al., 2017) based on a sum aggregation of messages that are obtained by relative position-dependent *linear* transformations of neighbouring node features. The power of message passing layers is, however, that node features are transformed and propagated in a highly *non-linear* manner. Equivariant GNNs have been proposed before as either PointConv-type (Wu et al., 2019; Kristof et al., 2017) implementations of steerable (Thomas et al., 2018; Anderson et al., 2019; Fuchs et al., 2020) or regular group convolutions (Finzi et al., 2020). The most important component in these methods are the convolution layers. Although powerful, such layers only (pseudo[1]) linearly transform the graphs and non-linearity is only obtained via point-wise activations.

---

[1]Methods such as SE(3)-transformers (Fuchs et al., 2020) and Cormorant (Anderson et al., 2019) include an input-dependent attention component that augments the convolutions.

In this paper, we propose non-linear E(3) *equivariant message passing* layers using the same principles that underlie steerable group convolutions, and view them as *non-linear group convolutions*. Central to our method is the use of steerable vectors and their equivariant transformations to represent and process node features; we present the underlying mathematics of both in Sec. 2 and illustrate it in Fig. 1 on a molecular graph. As a consequence, information at nodes and edges can now be rotationally invariant (scalar) or covariant (vector, tensor). In steerable message passing frameworks, the Clebsch-Gordan (CG) tensor product is used to steer the update and message functions by geometric information such as relative orientation (pose). Through a notion of steerable node attributes we provide a new class of equivariant activation functions for general use with steerable feature fields (Weiler et al., 2018; Thomas et al., 2018). Node attributes can include information such as node velocity, force, or atomic spin. Currently, especially in molecular modelling, most datasets are build up merely of atomic number and position information. In this paper, we demonstrate the potential of enriching node attributes with more geometric and physical quantities. We demonstrate the effectiveness of SEGNNs by setting a new state of the art on n-body toy datasets, in which our method leverages the abundance of geometric and physical quantities available. We further test our model on the molecular datasets QM9 and OC20. Although here only (relative) positional information is available as geometric quantity, SEGNNs achieve state of the art on the IS2RE dataset of OC20, and competitive performance on QM9. For all experiments we provide extensive ablation studies.

The main contributions of this paper are: (i) A generalisation of equivariant GNNs such that node and edge attributes are not restricted to scalars. (ii) A new class of equivariant activation functions for steerable vector fields, based on the introduction of steerable node attributes and steerable multi-layer perceptrons, which permit the injection of geometric and physical quantities into node updates. (iii) A unifying view on various equivariant GNNs through the definition of non-linear convolutions. (iv) Extensive experimental ablation studies that shows the benefit of steerable over non-steerable (invariant) message passing, and the benefit of non-linear over linear convolutions.

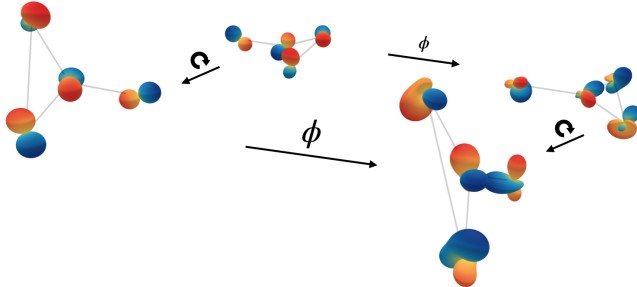

Figure 1: Commutation diagram for an equivariant operator $\phi$ applied to a 3D molecular graph with steerable node features (visualised as spherical functions); As the molecule rotates, so do the node features. The use of steerable vectors allows neural networks to exploit, embed, or learn geometric cues such as force and velocity vectors.

## 2 Generalised E(3) equivariant steerable message passing

**Message passing networks.**  Consider a graph $\mathcal{G} = (\mathcal{V}, \mathcal{E})$, with nodes $v_i \in \mathcal{V}$ and edges $e_{ij} \in \mathcal{E}$, with feature vectors $\mathbf{f}_i \in \mathbb{R}^{c_n}$ attached to each node, and edge attributes $\mathbf{a}_{ij} \in \mathbb{R}^{c_e}$ attached to each edge. Graph neural networks (GNNs) (Scarselli et al., 2009; Kipf & Welling, 2017; Defferrard et al., 2016; Battaglia et al., 2018) are designed to learn from graph-structured data and are by construction permutation equivariant with respect to the input. A specific type of GNNs are message passing networks (Gilmer et al., 2017), where a layer updates node features via the following steps:

$$\text{compute message } \mathbf{m}_{ij} \text{ from node } v_j \text{ to } v_i: \qquad \mathbf{m}_{ij} = \phi_m\left(\mathbf{f}_i, \mathbf{f}_j, \mathbf{a}_{ij}\right) , \qquad (1)$$

$$\text{aggregate messages and update node features } v_i: \qquad \mathbf{f}'_i = \phi_f\left(\mathbf{f}_i, \sum_{j \in \mathcal{N}(i)} \mathbf{m}_{ij}\right) , \qquad (2)$$

where $\mathcal{N}(i)$ represents the set of neighbours of node $v_i$, and $\phi_m$ and $\phi_f$ are commonly parameterised by multilayer perceptrons (MLPs).

**Equivariant message passing networks.** Our objective is to build graph neural networks that are robust to rotations, reflections, translations and permutations. This is a desirable property since some prediction tasks, such as molecular energy prediction, require E(3) invariance, whereas others, like force prediction, require equivariance. From a technical point of view, equivariance of a function $\phi$ to certain transformations means that for any transformation parameter $g$ and all inputs $x$ we have $T'_g[\phi(x)] = \phi(T_g[x])$, where $T_g$ and $T'_g$ denote transformations on the input and output domain of $\phi$, respectively. Equivariant operators applied to atomic graphs allow us to preserve the geometric structure of the system as well as enriching it with increasingly abstract directional information. We build E(3) equivariant GNNs by constraining the functions $\phi_m$ and $\phi_f$ of Eqs. (1-2) to be equivariant, which in return guarantees equivariance of the entire network. In the following, we introduce the core components behind our method; full mathematical details and background can be found in App. A.

**Steerable features.** In this work, we achieve equivariant graph neural networks by working with *steerable feature vectors*, which we denote with a tilde, e.g. a vector $\tilde{\mathbf{h}}$ is steerable. Steerability of a vector means that for a certain transformation group with transformation parameters $g$, the vector transforms via matrix-vector multiplication $\mathbf{D}(g)\tilde{\mathbf{h}}$. For example, a Euclidean vector in $\mathbb{R}^3$ is steerable for rotations $g = \mathbf{R} \in \mathrm{SO}(3)$ by multiplying the vector with a rotation matrix, thus $\mathbf{D}(g) = \mathbf{R}$. We are however not restricted to only work with 3D vectors; via the construction of steerable vector spaces, we can generalise the notion of 3D rotations to arbitrarily large vectors.

Central to our approach is the use of *Wigner-D matrices* $\mathbf{D}^{(l)}(g)$ [2]. These are $(2l + 1 \times 2l + 1)$-dimensional matrix representations that act on $(2l+1)$-dimensional vector spaces. These vector spaces that are transformed by $l$-th degree Wigner-D matrices will be referred to as *type-l steerable vector spaces* and denoted with $V_l$. We note that we can combine two independent steerable vector spaces $V_{l_1}$ and $V_{l_2}$ of type $l_1$ and $l_2$ by the direct sum, denoted by $V = V_{l_1} \oplus V_{l_2}$. Such a combined vector space then transforms by the direct sum of Wigner D-matrices, i.e., via $\mathbf{D}(g) = \mathbf{D}^{(l_1)}(g) \oplus \mathbf{D}^{(l_2)}(g)$, which is a block-diagonal matrix with the Wigner-D matrices along the diagonal. We denote the direct sum of type-$l$ vector spaces up to degree $l = L$ by $V_L := V_0 \oplus V_1 \oplus \cdots \oplus V_L$, and $n$ copies of the the same vector space with $nV := \underbrace{V \oplus V \oplus \ldots \oplus V}_{n \text{ times}}$. Regular MLPs are based on transformations between $d$-dimensional type-0 vector spaces i.e., $\mathbb{R}^d = dV_0$, and are a special case of our steerable MLPs that act on steerable vector spaces of arbitrary type.

**Steerable MLPs.** Like regular MLPs, steerable MLPs are constructed by interleaving linear mappings (matrix-vector multiplications) with non-linearities. Now however, the linear maps transform between steerable vector spaces at layer $i - 1$ to layer $i$ via $\tilde{\mathbf{h}}^i = \mathbf{W}^i_{\tilde{\mathbf{a}}}\tilde{\mathbf{h}}^{i-1}$. Steerable MLPs thus have the same functional form as regular MLPs, although, in our case, the linear transformation matrices $\mathbf{W}^i_{\tilde{\mathbf{a}}}$, defined below, are conditioned on geometric information (e.g. relative atom positions) which is encoded in the steerable vector $\tilde{\mathbf{a}}$. Both vectors $\tilde{\mathbf{a}}$ and $\tilde{\mathbf{h}}$ are steerable vectors. In this work we will however use the vector $\tilde{\mathbf{a}}$ to have geometric and structural information encoded and the steer the information flow of $\tilde{\mathbf{h}}$ through the network. In order to guarantee that $\mathbf{W}^i_{\tilde{\mathbf{a}}}$ maps between steerable vector spaces, the matrices are defined via the Clebsch-Gordan tensor product. By construction the resultant MLPs are equivariant for every transformation parameter $g$ via

$$\widetilde{\mathrm{MLP}}(\mathbf{D}(g)\tilde{\mathbf{h}}_0) = \mathbf{D}'(g)\widetilde{\mathrm{MLP}}(\tilde{\mathbf{h}}_0) \,, \tag{3}$$

provided that the steerable vectors $\tilde{\mathbf{a}}$ that condition the MLPs are also obtained equivariantly.

**Spherical harmonic embedding of vectors.** We can convert any vector $\mathbf{x} \in \mathbb{R}^3$ into a type-$l$ vector through the evaluation of *spherical harmonics* $Y_m^{(l)} : S^2 \to \mathbb{R}$ at $\frac{\mathbf{x}}{\|\mathbf{x}\|}$. For any $\mathbf{x} \in \mathbb{R}^3$

$$\tilde{\mathbf{a}}^{(l)} = \left(Y_m^{(l)}\left(\frac{\mathbf{x}}{\|\mathbf{x}\|}\right)\right)^T_{m=-l,-l+1,\ldots,l} \tag{4}$$

is a type-$l$ steerable vector. The spherical harmonic functions $Y_m^{(l)}$ are functions on the sphere $S^2$ and we visualise them as such in Figure 2. We will use spherical harmonic embeddings to include geometric and physical information into steerable MLPs.

**Mapping between steerable vector spaces.** The Clebsch-Gordan (CG) tensor product $\otimes_{cg} : V_{l_1} \times V_{l_2} \to V_l$ is a bilinear operator that combines two O(3) steerable input vectors of types $l_1$ and

---

[2]In order to be O(3)—not just SO(3)—equivariant, we include reflections. See App. A for more detail.

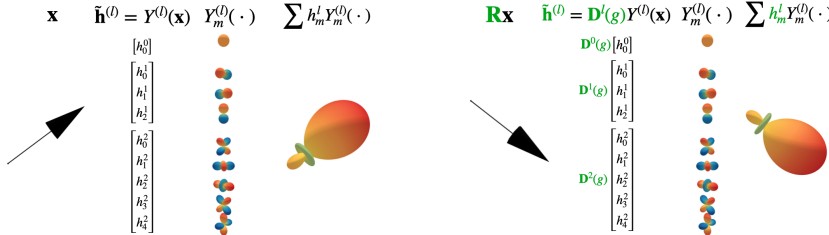

Figure 2: Left: representation of an O(3) steerable vector $\tilde{\mathbf{h}} \in V_L = V_0 \oplus V_1 \oplus V_2$, a spherical harmonic embedding of vector $\mathbf{x}$, e.g. relative orientation, velocity or force. Right: for each subspace the embedding with the basis functions $Y_m^{(l)}$ is shown (a). The transformation of $\tilde{\mathbf{h}}$ via $\mathbf{D}^{(l)}(g)$ acts on each subspace of $V_l$ separately (b).

$l_2$ and returns another steerable vector of type $l$. Let $\tilde{\mathbf{h}}^{(l)} \in V_l = \mathbb{R}^{2l+1}$ denote a steerable vector of type $l$ and $h_m^{(l)}$ its components with $m = -l, -l+1, \ldots, l$. The CG tensor product is given by

$$(\tilde{\mathbf{h}}^{(l_1)} \otimes_{cg}^w \tilde{\mathbf{h}}^{(l_2)})_m^{(l)} = w \sum_{m_1=-l_1}^{l_1} \sum_{m_2=-l_2}^{l_2} C_{(l_1,m_1)(l_2,m_2)}^{(l,m)} h_{m_1}^{(l_1)} h_{m_2}^{(l_2)} , \tag{5}$$

in which $w$ is a learnable parameter that scales the product, and $C_{(l_1,m_1)(l_2,m_2)}^{(l,m)}$ are the Clebsch-Gordan coefficients that ensure that the resulting vector is type-$l$ steerable. The CG tensor product is a sparse tensor product, as generally many coefficients are zero. Most notably, $C_{(l_1,m_1)(l_2,m_2)}^{(l,m)} = 0$ whenever $l < |l_1 - l_2|$ or $l > l_1 + l_2$. While Eq. (5) only describes the product between steerable vectors of a single type, e.g. $\tilde{\mathbf{h}}^{(l_1)} \in V_{l_1}$ and $\tilde{\mathbf{h}}^{(l_2)} \in V_{l_2}$, it is directly extendable to mixed type steerable vectors that may have multiple channels/multiplicities within a type. In this case, every input to output sub-vector pair gets its own index in a similar way as the weights in a standard linear layer are indexed with input-output indices. We then denote the CG product with $\otimes_{cg}^{\mathbf{W}}$ with boldfaced $\mathbf{W}$ to indicate that it is parametrised by a collection of weights. In order to stay close to standard notation used with MLPs, we treat the CG product with a fixed vector $\tilde{\mathbf{a}}$ in one of its inputs as a *steerable linear layer conditioned on* $\tilde{\mathbf{a}}$, denoted with

$$\mathbf{W}_{\tilde{\mathbf{a}}} \tilde{\mathbf{h}} := \tilde{\mathbf{h}} \otimes_{cg}^{\mathbf{W}} \tilde{\mathbf{a}} , \quad \text{or} \quad \mathbf{W}_{\tilde{\mathbf{a}}}(d) \tilde{\mathbf{h}} := \tilde{\mathbf{h}} \otimes_{cg}^{\mathbf{W}(d)} \tilde{\mathbf{a}} , \tag{6}$$

where the latter indicates that the CG weights depend on some quantity $d$, e.g. relative distances.

**Steerable activation functions.** The common recipe for deep neural networks is to alternate linear layers with element-wise non-linear activation functions. In the steerable setting, careful consideration is required to ensure that the activation functions are equivariant; currently available classes of activations include Fourier-based (Cohen et al., 2018), norm-altering (Thomas et al., 2018), or gated non-linearities (Weiler et al., 2018). We use gated non-linearities in all our architectures and shortly discuss their working principle in Sec. C in the appendix. The resulting steerable MLPs themselves in turn provide a new class of steerable activation functions, that is the first in its kind in directly leveraging local geometric cues. Namely, through steerable node attributes $\tilde{\mathbf{a}}$, either derived from the physical setup (forces, velocities) or from predictions (similar to gating). The MLPs can be applied node-wise and be generally used in steerable feature fields as non-linear activations.

## 2.1 STEERABLE E(3) EQUIVARIANT GRAPH NEURAL NETWORKS

We extend the message passing equations (1)-(2) and define a message passing layer that updates steerable node features $\tilde{\mathbf{f}}_i \in V_L$ at node $v_i$ via the following steps:

$$\tilde{\mathbf{m}}_{ij} = \phi_m \left( \tilde{\mathbf{f}}_i, \tilde{\mathbf{f}}_j, \|\mathbf{x}_j - \mathbf{x}_i\|^2, \tilde{\mathbf{a}}_{ij} \right) , \tag{7}$$

$$\tilde{\mathbf{f}}_i' = \phi_f \left( \tilde{\mathbf{f}}_i, \sum_{j \in \mathcal{N}(i)} \tilde{\mathbf{m}}_{ij}, \tilde{\mathbf{a}}_i \right) . \tag{8}$$

Here, $\|\mathbf{x}_j - \mathbf{x}_i\|^2$ is the squared relative distance between two nodes $v_i$ and $v_j$, $\phi_m$ and $\phi_f$ are O(3) steerable MLPs, $\tilde{\mathbf{a}}_{ij} \in V_L$ and $\tilde{\mathbf{a}}_i \in V_L$ are steerable edge and node attributes. If additional attributes exist, such as pair-wise distance $\|\mathbf{x}_j - \mathbf{x}_i\|$, they can either be concatenated to the attributes that condition our steerable MLPs, or as is more commonly done (Sec. 3) add as inputs to $\phi_m$ and $\phi_f$. We do the latter, and stack all inputs to a single steerable vector by which the steerable MLP layer is given: $\tilde{\mathbf{h}}^1 = \sigma(\mathbf{W}_{\tilde{\mathbf{a}}_{ij}} \tilde{\mathbf{h}}_i^0)$, with $\tilde{\mathbf{h}}_i^0 = \left( \tilde{\mathbf{f}}_i, \tilde{\mathbf{f}}_j, \|\mathbf{x}_j - \mathbf{x}_i\|^2 \right) \in V_f \oplus V_f \oplus V_0$, where $V_f$ is the user specified steerable vector space of node representations. The message network $\phi_m$ is steered via edge attributes $\tilde{\mathbf{a}}_{ij}$, and the node update network $\phi_f$ is similarly steered via node attributes $\tilde{\mathbf{a}}_i$.

**Injecting geometric and physical quantities.** In order to make SEGNNs more expressive, we include geometric and physical information in the edge and node updates. For that purpose, the edge attributes are obtained via the spherical harmonic embedding (Eq. (4)) of relative positions, in most cases, but possibly also relative force or relative momentum. The node attributes could e.g. be the average edge embedding of relative positions over neighbours of a node, i.e., $\tilde{\mathbf{a}}_i = \frac{1}{|\mathcal{N}(i)|} \sum_{j \in \mathcal{N}(i)} \tilde{\mathbf{a}}_{ij}$, and could additionally include node force, spin or velocities, as we do in the N-body experiment. The use of steerable node attributes in the steerable MLPs that define $\phi_f$ allows us to not just integrate geometric cues into the message functions, but also leverage it in the node updates. We observe that the more geometric and physical quantities are injected the better SEGNNs perform.

## 3 MESSAGE PASSING AS CONVOLUTION, RELATED WORK

Recent literature shows a trend towards building architectures that improve performance by means of maximally preserving equivariance through groups convolutions (either in regular or tensor-product form, see App. B). Convolutions, however, are "just" linear operators and non-linearities are only introduced through point-wise activation functions. This is in contrast to architectures that are built without explicit use of group convolutions, but instead rely on the highly non-linear framework of message passing. In the following we show that many related works are connected through a notion of *non-linear convolution*, a term that we coin based on the following. Any linear operator which is equivariant is a group convolution (Kondor & Trivedi, 2018; Bekkers, 2019) and their discrete implementations can be written in message passing form. We then call any non-linear operator that is equivariant, and which can be written in simple message passing form, a *non-linear convolution*. This framing allows us to place related work in a unifying context and to identify two important aspects of successful architectures: (i) equivariant layers improve upon invariant ones and (ii) non-linear layers improve upon linear ones. Both come together in SEGNNs via steerable non-linear convolutions.

**Point convolutions as equivariant linear message passing.** Consider a feature map $\mathbf{f} : \mathbb{R}^d \to \mathbb{R}^{c_l}$. A convolution layer (defined via cross-correlation) with a point-wise non-linearity $\sigma$ is given by

$$\mathbf{f}'(\mathbf{x}) = \sigma \left( \int_{\mathbb{R}^d} \mathbf{W}(\mathbf{x}' - \mathbf{x}) \mathbf{f}(\mathbf{x}') \mathrm{d}\mathbf{x}' \right), \tag{9}$$

with $\mathbf{W} : \mathbb{R}^d \to \mathbb{R}^{c_{l+1} \times c_l}$ a convolution kernel that provides for every relative position a matrix that linearly transforms features from $\mathbb{R}^{c_l}$ to $\mathbb{R}^{c_{l+1}}$. *Point convolutions*, generally referred to as PointConvs (Wu et al., 2019), and SchNet (Kristof et al., 2017) implement Eq. (9) on point clouds. For a sparse input feature map consisting of location-feature pairs $(\mathbf{x}_i, \mathbf{f}_i)$, the point convolution is given by $\mathbf{f}_i' = \sum_{j \in \mathcal{N}(i)} \mathbf{W}(\mathbf{x}_j - \mathbf{x}_i) \mathbf{f}_i$, which describes a *linear message passing layer* of Eqs. (1)-(2) in which the messages are $\mathbf{m}_{ij} = \mathbf{W}(\mathbf{x}_j - \mathbf{x}_i) \mathbf{f}_j$ and the message update $\mathbf{f}_i' = \sum_j \mathbf{m}_{ij}$. In the above convolutions, the transformation matrices $\mathbf{W}$ are conditioned on relative positions $\mathbf{x}_j - \mathbf{x}_i$, which is typically done in one of the following three approaches. (i) Classically, feature maps are processed on dense regular grids with shared neighbourhoods and thus the transformations $\mathbf{W}_{ij}$ can be stored for a finite set of relative positions $\mathbf{x}_j - \mathbf{x}_i$ in a single tensor. This method however does not generalise to non-uniform grids such as point clouds. Continuous kernel methods parametrise the transformations either by (ii) expanding $\mathbf{W}$ into a continuous basis or (iii) directly parametrising them with MLPs via $\mathbf{W}(\mathbf{x}_j - \mathbf{x}_i) = \mathrm{MLP}(\mathbf{x}_j - \mathbf{x}_i)$. Steerable kernel methods[3] are of type (ii) and rely on a *steerable*

---

[3] See (Lang & Weiler, 2020) for a general theory for $G$-steerable kernel constraints for compact groups.

*basis*, such as 3D spherical harmonics $Y_m^{(l)}$, via

$$\mathbf{W}(\mathbf{x}_j - \mathbf{x}_i) = \sum_l \sum_{m=-l}^l \mathbf{W}_m^{(l)}(\|\mathbf{x}_j - \mathbf{x}_i\|) Y_m^{(l)}(\mathbf{x}_j - \mathbf{x}_i) , \tag{10}$$

with basis coefficients $\mathbf{W}_m^{(l)}$ typically depending on pair-wise distances. We next show how such kernels connect to steerable group convolutions and provide a detailed background in App. B.

**Steerable (group) convolutions.** In context of the steerable framework of Sec. 2, linear feature transformations $\mathbf{W}$ are equivalent to steerable linear transformations (Eq. (6)) conditioned on the scalar "1". I.e, with $\mathbf{h}_i \in \mathbb{R}^d$ in the usual and $\tilde{\mathbf{h}} \in dV_0$ the steerable setting, messages are obtained by

$$\mathbf{m}_{ij} = \mathbf{W}(\mathbf{x}_j - \mathbf{x}_i)\mathbf{h}_i \quad \Leftrightarrow \quad \tilde{\mathbf{m}}_{ij} = \mathbf{W}_1(\mathbf{x}_j - \mathbf{x}_i)\tilde{\mathbf{h}}_i . \tag{11}$$

When the transformations are parametrised in a steerable basis (10) we can make the identification[4]

$$\mathbf{m}_{ij} = \mathbf{W}(\mathbf{x}_j - \mathbf{x}_i)\mathbf{h}_i \quad \Leftrightarrow \quad \tilde{\mathbf{m}}_{ij} = \mathbf{W}_{\tilde{\mathbf{a}}_{ij}}(\|\mathbf{x}_j - \mathbf{x}_i\|)\tilde{\mathbf{h}}_i , \tag{12}$$

in which $\tilde{\mathbf{a}}_{ij} = Y_m^{(l)}(\mathbf{x}_j - \mathbf{x}_i)$ are spherical harmonic embeddings (Eq. (4)) of $\mathbf{x}_j - \mathbf{x}_i$, and the weights that parametrise the CG tensor product depend on pair-wise distances. We can thus perform convolutions either in the regular or in the steerable setting, where the latter has the benefit of allowing us to directly derive what the convolution result would be if the kernel were to be rotated via

$$\mathbf{m}_{ij} = \mathbf{W}(\mathbf{R}^{-1}(\mathbf{x}_j - \mathbf{x}_i))\mathbf{h}_i \quad \Leftrightarrow \quad \tilde{\mathbf{m}}_{ij} = \mathbf{D}(\mathbf{R})\mathbf{W}_{\tilde{\mathbf{a}}_{ij}}(\|\mathbf{x}_j - \mathbf{x}_i\|)\tilde{\mathbf{h}}_i . \tag{13}$$

Steerable vectors obtained via convolutions with steerable kernels thus generate signals on O(3) via the Wigner-D matrices. This relation is in fact established by the inverse Fourier transform on O(3) (cf. App. A), by which we can treat steerable feature vectors $\tilde{\mathbf{f}}_i$ at each location $\mathbf{x}_i$ as a function on the group O(3). Steerable convolutions thus produce feature maps on the full group E(3) that for every possible translation/position $\mathbf{x}$ and rotation $\mathbf{R}$ provide a feature response $\mathbf{f}(\mathbf{x}, \mathbf{R})$. It is precisely this mechanism of transforming convolution kernels via the group action (via $\mathbf{W}(g^{-1} \cdot \mathbf{x}_j) = \mathbf{W}(\mathbf{R}^{-1}(\mathbf{x}_j - \mathbf{x}))$) that underlies group convolutions (Cohen & Welling, 2016). Message passing via explicit kernel rotations (l.h.s. of (13)) corresponds to *regular group convolutions*, and via steerable transformations (r.h.s. of (13)) to *steerable group convolutions*.

The equivariant steerable methods (Thomas et al., 2018; Anderson et al., 2019; Miller et al., 2020; Fuchs et al., 2020) that we compare against in our experiments can all be written in convolution form

$$\tilde{\mathbf{f}}_i' = \sum_{j \in \mathcal{N}(i)} \mathbf{W}_{\tilde{\mathbf{a}}_{ij}}(\|\mathbf{x}_j - \mathbf{x}_i\|)\tilde{\mathbf{f}}_j , \qquad \text{or} \qquad \tilde{\mathbf{f}}_i' = \sum_{j \in \mathcal{N}(i)} \mathbf{W}_{\tilde{\mathbf{a}}_{ij}}(\tilde{\mathbf{f}}_i, \tilde{\mathbf{f}}_j, \|\mathbf{x}_j - \mathbf{x}_i\|)\tilde{\mathbf{f}}_j , \tag{14}$$

where, in the latter case, the linear transformations additionally depend on an input dependent attention mechanism as in (Anderson et al., 2019; Fuchs et al., 2020), and can be seen as a steerable PointConv version of attentive group convolutions (Romero et al., 2020). In these attention-based cases, convolutions are augmented with input dependent weights $\alpha$ via $\mathbf{W}_{\tilde{\mathbf{a}}_{ij}}(\tilde{\mathbf{f}}_i, \tilde{\mathbf{f}}_j, \|\mathbf{x}_j - \mathbf{x}_i\|) = \alpha(\tilde{\mathbf{f}}_i, \tilde{\mathbf{f}}_j)\mathbf{W}_{\tilde{\mathbf{a}}_{ij}}(\|\mathbf{x}_j - \mathbf{x}_i\|)$. This makes the convolution non-linear, however, the transformation of input features still happens linearly and thus describes what one may call a pseudo-linear transformation. Finally, the recently proposed LieConv (Finzi et al., 2020) and NequIP (Batzner et al., 2021) also fall in the convolutional message passing class. LieConv is a PointConv-type variation of *regular group convolutions* on Lie groups (Bekkers, 2019). NeuqIP follows the approach of Tensor Field Networks (Thomas et al., 2018), and weighs interactions using an MLP with radial basis functions as input. These functions are obtained as solution of the Schrödinger equation. Finally, steerable methods fall into a more general class of coordinate independent convolutions (Weiler et al., 2021) which allow to ensure equivariance locally, even when global symmetries can not be defined.

**Equivariant message passing as non-linear convolution.** EGNNs (Satorras et al., 2021) are equivariant to transformations in E($n$) and outperform most aforementioned steerable methods. This is somewhat surprising as it sends *invariant* messages, which are obtained via MLPs of the form

$$\mathbf{m}_{ij} = \text{MLP}(\mathbf{f}_i, \mathbf{f}_j, \|\mathbf{x}_j - \mathbf{x}_i\|^2) = \sigma(\mathbf{W}^{(k)}(\ldots(\sigma(\mathbf{W}^{(1)}\mathbf{h}_i)))) , \tag{15}$$

---

[4]Exact correspondence is obtained by a sum reduction over the steerable vector components (App. B).

where $\mathbf{h}_i = (\mathbf{f}_i, \mathbf{f}_j, \|\mathbf{x}_j - \mathbf{x}_i\|^2)$. These messages resemble the convolutional messages of point convolutions due to their dependency on relative positions. There are, however, two important differences: (i) the messages are non-linear transformations of the neighbouring feature values $\mathbf{f}_j$ via an MLP and (ii) the messages are only conditioned on the distance between point pairs, and are therefore E($n$) invariant. As such, we regard EGNN layers as *non-linear convolutions* with *isotropic message* functions (the non-linear counterpart of rotationally invariant kernels). In our work, we lift the isotropy constraint and generalise to non-linear steerable convolutions via messages of the form

$$\tilde{\mathbf{m}}_{ij} = \widetilde{\mathrm{MLP}}_{\tilde{\mathbf{a}}_{ij}}(\tilde{\mathbf{f}}_i, \tilde{\mathbf{f}}_j, \|\mathbf{x}_j - \mathbf{x}_i\|^2) = \sigma(\mathbf{W}^{(n)}_{\tilde{\mathbf{a}}_{ij}}(\ldots(\sigma(\mathbf{W}^{(1)}_{\tilde{\mathbf{a}}_{ij}}\tilde{\mathbf{h}}_i)))) \,, \tag{16}$$

with $\tilde{\mathbf{h}}_i = (\tilde{\mathbf{f}}_i, \tilde{\mathbf{f}}_j, \|\mathbf{x}_j - \mathbf{x}_i\|^2) \in V_f \oplus V_f \oplus V_0$. The MLP is then conditioned on attribute $\tilde{\mathbf{a}}_{ij}$, which could e.g. be a spherical harmonic embedding of $\mathbf{x}_j - \mathbf{x}_i$. This allows for the creation of messages more general than those found in convolution, while carrying covariant geometrical information.

**Related equivariant message passing methods.** A different but also fully message passing based approach can be found in Geometric Vector Perceptrons (GVP) (Jing et al., 2020), Vector Neurons (Deng et al., 2021), and PaiNN (Schütt et al., 2021). Compared to SEGNNs which treat equivariant information as fully steerable features, these architectures update scalar-valued attributes using the norm of vector-valued attributes, and therefore with O(3) invariant information. These methods restrict the flow of information between attributes of different types, whereas the Clebsch-Gordan tensor product in SEGNNs allows for interaction between spherical harmonics of all orders throughout the network. Methods such as Dimenet++ (Klicpera et al., 2020), SphereNet (Liu et al., 2021), and GemNet (Klicpera et al., 2021) incorporate relative orientation in a *second-order* message passing scheme that considers angles between the central point and neighbours-of-neighbours. In contrast, our method directly leverages angular information in a *first-order* message passing scheme using steerable vectors. We attribute the success of such methods to the fact that they equivariantly process point clouds of local orientations $(\mathbf{x}_i, \mathbf{r}_{ij}) \in \mathbb{R}^3 \times S^2$, defined by relative positions between atoms, by sending messages between edges (local orientations) rather than nodes. As such, they can be thought of as non-linear *regular group convolutions* on the homogeneous space of positions and orientations ($\mathbb{R}^3 \times S^2$) with isotropic (zonal) message functions, where the symmetry constraint is induced by the quotient $\mathbb{R}^3 \times S^2 \equiv \mathrm{SE}(3)/\mathrm{SO}(2)$ (Bekkers, 2019, Thm. 1).

## 4 EXPERIMENTS

**Implementation details.** The implementation of SEGNN's O(3) steerable MLPs is based on the `e3nn` library (Geiger et al., 2021a). We either define the steerable vector spaces as $V = nV_{L=l_{max}}$ (N-body, QM9 experiments), i.e., $n$ copies of steerable vector spaces up to order $l_{max}$, or by dividing an $n$-dimensional vector space $V$ into $L$ approximately equally large type-$l$ sub-vector spaces (OC20 experiments) as is done in Finzi et al. (2021). Furthermore, for a fair comparison between experiments with different $l_{max}$, we choose $n$ such that the total number of weights in the CG products corresponds to that of a regular (type-0) linear layer. Further implementation details are in App. C.

**SEGNN architectures and ablations.** We consider several variations of SEGNNs. On all tasks we have at least one fully steerable ($l_f > 0, l_a > 0$) SEGNN tuned for the specific task at hand. We perform ablation experiments to investigate two main principles that sets SEGNNs apart from the literature. **A1** The case of non-steerable vs steerable EGNNs is obtained by applying the same SEGNN network with different specifications of maximal spherical harmonic order in the feature vectors ($l_f$) and the attributes ($l_a$). EGNN (Satorras et al., 2021) arises as a special case with $l_f = l_a = 0$. These models will be labelled SEGNN. **A2** In this ablation, we use steerable equivariant point conv methods (Thomas et al., 2018) with messages as in Eq. (14) and regular gated non-linearities as activation/update function, labelled SE$_{\mathrm{linear}}$, and compare it to the same network but with messages obtained in a non-linear manner via 2-layer steerable MLPs as in Eq. (16), labelled as SE$_{\mathrm{non-linear}}$.

**N-body system.** The charged N-body particle system experiment (Kipf et al., 2018) consists of 5 particles that carry a positive or negative charge, having initial position and velocity in a 3-dimensional space. The task is to estimate all particle positions after 1.000 timesteps. We build upon the experimental setting introduced in (Satorras et al., 2021). Steerable architectures are designed such that the parameter budget at $l_f = 1$ and $l_a = 1$ matches that of the EGNN implementation. We input the relative position to the center of the system and the velocity as vectors of type $l = 1$ with odd parity. We further input the norm of the velocity as scalar, which altogether results in an input

Table 1: Mean Squared Error (MSE) in the N-body system experiment, and forward time in seconds for a batch size of 100 samples running on a GeForce RTX 2080 Ti GPU. Results except for EGNN and SEGNN are taken from (Satorras et al., 2021) and verified. Runtimes are re-measured.

| Method | MSE | Time [s] |
|---|---|---|
| SE(3)-Tr. (Fuchs et al., 2020) | .0244 | .0742 |
| TFN (Thomas et al., 2018) | .0155 | .0182 |
| NMP (Gilmer et al., 2017) | .0107 | .0017 |
| Radial Field (Köhler et al., 2019) | .0104 | .0019 |
| EGNN (Satorras et al., 2021) | $.0070 \pm .00022$ | .0029 |
| $SE_{linear}$ ($l_f = 2, l_a = 2$) | $.0116 \pm .00021$ | .0640 |
| $SE_{non\text{-}linear}$ ($l_f = 1, l_a = 1$) | $.0060 \pm .00019$ | .0310 |
| $SEGNN_G$ ($l_f = 1, l_a = 1$) | $.0056 \pm .00025$ | .0250 |
| $SEGNN_{G+P}$ ($l_f = 1, l_a = 1$) | $.0043 \pm .00015$ | .0260 |

vector $\tilde{\mathbf{h}} \in V_0 \oplus V_1 \oplus V_1$. The output is embedded as difference vector to the initial position (odd parity), i.e. $\tilde{\mathbf{o}} \in V_1$. In doing so, we keep E(3) equivariance for vector valued inputs and outputs. The edge attributes are obtained via the spherical harmonic embedding of $\mathbf{x}_j - \mathbf{x}_i$ as described in Eq. 4. Messages additionally have the product of charges and absolute distance included. SEGNN architectures are compared to steerable equivariant ($SE_{linear}$) and steerable non-linear point conv methods ($SE_{non\text{-}linear}$). Results and ablation studies are shown in Tab. 1. A full ablation is outlined in App. C. Steerable architectures obtain the best results for $l_f = 1$ and $l_a = 1$, and don't benefit from higher orders of $l_f$ and $l_a$. We consider a first SEGNN architecture where the node attributes are the averaged edge embeddings, i.e. mean over relative orientation, labelled $SEGNN_G$ since only geometric information is used. The second SEGNN architecture has the spherical harmonics embedding of the velocity added to the node attributes. It can thus leverage geometric (orientation) and physical (velocity) information, and is consequently labelled $SEGNN_{G+P}$. Including physical information in addition to geometric information in the node updates considerably boosts SEGNN performance. We further test SEGNNs on a gravitational 100-body system (Sec. C.2 in the appendix).

**QM9.** The QM9 dataset (Ramakrishnan et al., 2014; Ruddigkeit et al., 2012) consists of small molecules up to 29 atoms, where each atom is described with 3D position coordinates and one-hot mode embedding of its atomic type (H, C, N, O, F). The aim is to regress various chemical properties for each of the molecules, optimising on the mean absolute error (MAE) between predictions and ground truth. We use the dataset partitions from Anderson et al. (2019). Table 2 shows SEGNN results on the QM9 dataset. In Table 3, we show that by steering with the relative orientation between atoms we observe that for higher (maximum) orders of steerable feature vectors, the performance increases, especially when a small cutoff radius is chosen. While previous methods use relatively large cutoff radii of 4.5-11Å, we use a cutoff radius of 2Å. Doing so results in a sharp reduction of the number of messages per layer, as shown in App. C. Tables 2 and 3 together show that SEGNNs outperform an architecturally comparable baseline EGNN, whilst stripping away attention modules from it and reducing graph connectivity from fully connected to only 2Å distant atoms. It is however apt to note that runtime is still limited by the relatively expensive calculation of the Clebsch-Gordan tensor products. We further note that SEGNNs produce results on par with the best performing methods on the non-energy variables, however lag behind state of the art on the energy variables ($G$, $H$, $U$, $U_0$). We conjecture that such targets could benefit from more involved (e.g. including attention or neighbour-neighbour interactions) or problem-tailored architectures, such as those compared against.

**OC20.** The Open Catalyst Project OC20 dataset (Zitnick et al., 2020; Chanussot et al., 2021), consists of molecular adsorptions onto surfaces. We focus on the Initial Structure to Relaxed Energy (IS2RE) task, which takes as input an initial structure and targets the prediction of the energy in the final, relaxed state. The IS2RE training set consists of over 450,000 catalyst adsorbate combinations with 70 atoms on average. Optimisation is done for MAE between the predicted and ground truth energy. Additionally, performance is measured in the percentage of structures in which the predicted energy is within a $0.02$ eV threshold (EwT). The four test splits contain in-distribution (ID) catalysts and adsorbates, out-of-domain adsorbates (OOD Ads), out-of-distribution catalysts (OOD Cat), and out-of-distribution adsorbates and catalysts (OOD Both). Table 4 shows SEGNN results on the OC20 dataset and comparisons with existing methods. We compare to models which have obtained results by training on the IS2RE training set such as SphereNet (Liu et al., 2021) and DimeNet++ (Klicpera

et al., 2019; 2020). The best SEGNN performance is seen for $l_f = 1$ and $l_a = 1$ (see App. C). A full ablation study, comparing performance and runtime for different orders $l_f$ and $l_a$ is found in App. C).

Table 2: Performance comparison on the QM9 dataset. Numbers are reported for Mean Absolute Error (MAE) between model predictions and ground truth.

| Task | $\alpha$ | $\Delta\varepsilon$ | $\varepsilon_{\mathrm{HOMO}}$ | $\varepsilon_{\mathrm{LUMO}}$ | $\mu$ | $C_\nu$ | $G$ | $H$ | $R^2$ | $U$ | $U_0$ | ZPVE |
| Units | bohr$^3$ | meV | meV | meV | D | cal/mol K | meV | meV | bohr$^3$ | meV | meV | meV |
| NMP | .092 | 69 | 43 | 38 | .030 | .040 | 19 | 17 | .180 | 20 | 20 | 1.50 |
| SchNet * | .235 | 63 | 41 | 34 | .033 | .033 | 14 | 14 | .073 | 19 | 14 | 1.70 |
| Cormorant | .085 | 61 | 34 | 38 | .038 | .026 | 20 | 21 | .961 | 21 | 22 | 2.02 |
| L1Net | .088 | 68 | 46 | 35 | .043 | .031 | 14 | 14 | .354 | 14 | 13 | 1.56 |
| LieConv | .084 | 49 | 30 | 25 | .032 | .038 | 22 | 24 | .800 | 19 | 19 | 2.28 |
| TFN | .223 | 58 | 40 | 38 | .064 | .101 | - | - | - | - | - | - |
| SE(3)-Tr. | .142 | 53 | 35 | 33 | .051 | .054 | - | - | - | - | - | - |
| DimeNet++ * | **.043** | **32** | 24 | 19 | .029 | .023 | **7** | **6** | .331 | 6 | 6 | 1.21 |
| SphereNet * | .046 | **32** | **23** | **18** | .026 | **.021** | 8 | **6** | .292 | 7 | 6 | **1.12** |
| PaiNN * | .045 | 45 | 27 | 20 | **.012** | .024 | **7** | **6** | **.066** | **5** | **5** | 1.28 |
| EGNN | .071 | 48 | 29 | 25 | .029 | .031 | 12 | 12 | .106 | 12 | 12 | 1.55 |
| SEGNN (Ours) | .060 | 42 | 24 | 21 | .023 | .031 | 15 | 16 | .660 | 13 | 15 | 1.62 |

\* these methods use different train/val/test partitions.

Table 3: QM9 ablation study to compare SEGNN performances for different (maximum) orders of steerable feature vectors ($l_f$) and attributes ($l_a$). Models with only trivial features are akin to the invariant EGNN. The method with a fully connected graph uses soft edge estimation (Satorras et al., 2021). Forward time is measured for a batch of 128 samples running on a GeForce RTX 3090 GPU.

| Task | | $\alpha$ | $\Delta\varepsilon$ | $\varepsilon_{\mathrm{HOMO}}$ | $\varepsilon_{\mathrm{LUMO}}$ | $\mu$ | $C_\nu$ | |
| Units | Cutoff radius | bohr$^3$ | meV | meV | meV | D | cal/mol K | Time [s] |
| (S)EGNN ($l_f = 0, l_a = 0$) | - | .091 | 53 | 34 | 28 | .042 | .043 | 0.016 |
| (S)EGNN ($l_f = 0, l_a = 0$) | 5Å | .105 | 57 | 36 | 31 | .047 | .047 | 0.014 |
| SEGNN ($l_f = 1, l_a = 1$) | 5Å | .080 | 49 | 29 | 25 | .032 | .034 | 0.058 |
| (S)EGNN ($l_f = 0, l_a = 0$) | 2Å | .240 | 98 | 60 | 60 | .340 | .077 | 0.014 |
| SEGNN ($l_f = 1, l_a = 2$) | 2Å | .074 | 48 | 27 | 25 | .031 | .035 | 0.046 |
| SEGNN ($l_f = 2, l_a = 3$) | 2Å | .060 | 42 | 24 | 21 | .023 | .031 | 0.096 |

Table 4: Comparison on the OC20 IS2RE task in terms of Mean Absolute Error (MAE) between model predictions and ground truth energy and % of predictions within $\epsilon = 0.02$ eV of the ground truth (EwT). Numbers are taken from the OC20 leaderboard. SEGNNs outperform all competitors.

| | Energy MAE [eV] ↓ | | | | EwT ↑ | | | |
| Model | ID | OOD Ads | OODCat | OOD Both | ID | OOD Ads | OOD Cat | OOD Both |
| Median baseline | 1.7499 | 1.8793 | 1.7090 | 1.6636 | 0.71% | 0.72% | 0.89% | 0.74% |
| CGCNN | 0.6149 | 0.9155 | 0.6219 | 0.8511 | 3.40% | 1.93% | 3.10% | 2.00% |
| SchNet | 0.6387 | 0.7342 | 0.6616 | 0.7037 | 2.96% | 2.33% | 2.94% | 2.21% |
| EdgeUpdateNet | 0.5839 | 0.7252 | 0.6016 | 0.6862 | 3.48% | 2.35% | 3.30% | 2.57% |
| EnergyNet | 0.6366 | 0.717 | 0.6387 | 0.6626 | 3.30% | 2.20% | 3.07% | 2.34% |
| DimeNet++ | 0.5620 | 0.7252 | 0.5756 | 0.6613 | 4.25% | 2.07% | 4.10% | 2.41% |
| SphereNet | 0.5630 | 0.7030 | 0.5710 | **0.6380** | 4.47% | 2.29% | 4.09% | 2.41% |
| SEGNN (Ours) | **0.5327** | **0.6921** | **0.5369** | 0.6790 | **5.37%** | **2.46%** | **4.91%** | **2.63%** |

## 5 CONCLUSION

We have introduced SEGNNs which generalise equivariant graph networks, such that node and edge information is not restricted to be invariant (scalar), but can also be vector- or tensor-valued. SEGNNs are the first networks which allow the steering of node updates by leveraging geometric and physical cues, introducing a new class of equivariant activation functions. We demonstrate the potential of SEGNNs by applying it to a wide range of different tasks. Extensive ablation studies have further shown the benefit of steerable over non-steerable (invariant) message passing, and the benefit of non-linear over linear convolutions. On the OC20 ISRE taks, SEGNNs outperform all competitors.

## 6 REPRODUCIBILITY STATEMENT

We have included error bars, reproducibility checks and ablation studies wherever we found it necessary and appropriate. For example, for the N-body experiment we have reproduced the results of Satorras et al. (2021), we have ablated different (maximum) orders of steerable feature vectors, and we have stated mean and standard deviation of the results which we have obtained by running the same experiments eight times with different initial seeds. For the OC20 experiments, we have stated official numbers from the Open Catalyst Project challenge in the paper. Thus, our comparisons to other methods are obtained from their respective best entries to the challenge. We have further ablated runtimes, different cutoff radii and different (maximum) orders of steearable feature vectors, see Sec. C in the appendix. We have done this by running the experiments eight times with different initial random seeds to be able to report mean and standard deviation of the results. For the reproducibility of the QM9 experiments, we have uploaded our code and included a command which reproduces results of the $\alpha$ variable.

We have described our architecture in Sec. 2.1 and provided further implementation details in Appendix Section C. We have not introduced new mathematical results. However, we have used concepts from different mathematical fields and therefore included a self-contained mathematical exposition in our appendix. We have further included proof of equivariance and properties of Wigner D matrices and spherical harmonics in the appendix. In Sec. 3, we have introduced a unifying view on various equivariant graph neural networks through the definition of non-linear convolutions, which allows us to draw comparisons with many other methods; we verify our findings in the N-body experiments of Sec. 4.

## 7 ETHICAL STATEMENT

The societal impact of SEGNNs is difficult to predict. However, SEGNNs are well suited for physical and chemical modeling and therefore potential shortcuts for computationally expensive simulations. And if used as such, SEGNNs might potentially be directly or indirectly related to reducing the carbon footprint. On the downside, relying on simulations always requires rigorous cross-checks and monitoring, especially when simulations or simulated quantities are learned.

## ACKNOWLEDGMENTS

Johannes Brandstetter thanks the Institute of Advanced Research in Artificial Intelligence (IARAI) and the Federal State Upper Austria for the support. This work is part the research programme VENI (grant number 17290), financed by the Dutch Research Council (NWO). The authors thank Markus Holzleitner for helpful comments on this work.

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

# A  MATHEMATICAL BACKGROUND

This appendix provides the mathematical background and intuition for steerable MLPs. We remark that the reader may appreciate several related works, such as, (Thomas et al., 2018; Anderson et al., 2019; Fuchs et al., 2020), as excellent alternative resources[5] to get acquainted with the group/representation theory used in this paper. In this appendix we introduce the theory from our own perspective which is tuned towards the idea of steerable MLPs and our viewpoint on group convolutions. It provides complementary intuition to the aforementioned resources. The main concepts explained in this appendix are:

1. *Group definition* and *examples of groups* (Section A.1). The entire framework builds upon notions from group theory and as such a formal definition is in order. In this paper, we model transformations such as translation, rotation and reflection as groups.

2. *Invariance*, *equivariance* and *representations* (Section A.2). A function is said to be invariant to a transformation if its output is unaffected by a transformation of the input. A function is said to be equivariant if its output transforms predictably under a transformation of the input. In order to make the definition precise, we need a definition of representations; a representation formalises the notion of transformations applied to vectors in the context of group theory.

3. *Steerable vectors*, *Wigner-D matrices* and *irreducible representations* (Section A.3). Whereas regular MLPs work with feature vectors whose elements are scalars, our steerable MLPs work with feature vectors consisting of steerable feature vectors. Steerable feature vectors are vectors that transform via so-called Wigner-D matrices, which are representations of the orthogonal group O(3). Wigner-D matrices are the smallest possible group representations and can be used to define any representation (or conversely, any representation can be reduced to a tensor product of Wigner-D matrices via a change of basis). As such, the Wigner-D matrices are irreducible representations.

4. *Spherical harmonics* (Section A.4). Spherical harmonics are a class of functions on the sphere $S^2$ and can be thought of as a Fourier basis on the sphere. We show that spherical harmonics are steered by the Wigner-D matrices and interpret steerable vectors as functions on $S^2$, which justifies the glyph visualisations used in this paper. Moreover, spherical harmonics allow the embedding of three-dimensional displacement vectors into arbitrarily large steerable vectors.

5. *Clebsch-Gordan tensor product* and *steerable MLPs* (Section A.5). In a regular MLP one maps between input and output vector spaces linearly via matrix vector multiplication and applies non-linearities afterwards. In steerable MLPs one maps between steerable input and steerable output vector spaces via the Clebsch-Gordan tensor product. Akin to the learnable weight matrix in regular MLPs, the learnable Glebsch-Gordan tensor product is the workhorse of our steerable MLPs.

After these concepts are introduced we will in Section B revisit the convolution operator in the light of the steerable, group theoretical viewpoint that we take in this paper. In particular, we show that steerable group convolutions are equivalent to linear group convolutions with convolution kernels expressed in a spherical harmonic basis. With this in mind, we argue that our approach via message passing can be thought of as building neural networks via non-linear group convolutions.

## A.1  GROUP DEFINITION AND THE GROUPS E(3) AND O(3)

**Group definition.**  A group is an algebraic structure that consists of a set $G$ and a binary operator $\cdot$, the group product, that satisfies the following axioms: *Closure*: for all $h, g \in G$ we have $h \cdot g \in G$; *Identity*: there exists an identity element $e \in G$; *Inverse*: for each $g \in G$ there exists an inverse element $g^{-1} \in G$ such that $g^{-1} \cdot g = g \cdot g^{-1} = e$; and *Associativity*: for each $g, h, i \in G$ we have $(g \cdot h) \cdot i = g \cdot (h \cdot i)$.

**The Euclidean group E(3).**  In this work, we are interested in the group of three-dimensional translations, rotations, and reflections which is denoted with E(3), the 3D Euclidean group. Such

---

[5]Each of these works presents unique view points that greatly influenced the writing of this appendix.

transformations are parametrised by pairs of translation vectors $\mathbf{x} \in \mathbb{R}^3$ and orthogonal transformation matrices $\mathbf{R} \in \mathrm{O}(3)$. The E(3) group product and inverse are defined by

$$g \cdot g' := \left(\mathbf{R}\mathbf{x}' + \mathbf{x}, \mathbf{R}\mathbf{R}'\right),$$
$$g^{-1} := \left(\mathbf{R}^{-1}\mathbf{x}, \mathbf{R}^{-1}\right),$$

with $g = (\mathbf{x}, \mathbf{R}), g' = (\mathbf{x}', \mathbf{R}) \in \mathrm{E}(3)$. One can readily see that with these definitions all four group axioms are satisfied, and that it therefore defines a group. The group product can be seen as a description for how two E(3) transformations parametrised by $g$ and $g'$ applied one after another are described by single transformation parametrised by $g \cdot g'$. The transformations themselves act on the vector space of 3D positions via the group action, which we also denote with $\cdot$, via

$$g \cdot \mathbf{y} := \mathbf{R}\mathbf{y} + \mathbf{x},$$

where $g = (\mathbf{x}, \mathbf{R}) \in \mathrm{E}(3)$ and $\mathbf{y} \in \mathbb{R}^3$.

**The orthogonal group O$(3)$ and special orthogonal group SO$(3)$.** The group $\mathrm{E}(3) = \mathbb{R}^3 \rtimes \mathrm{O}(3)$ is a semi-direct product (denoted with $\rtimes$) of the group of translations $\mathbb{R}^3$ with the group of orthogonal transformations O(3). This means that we can conveniently decompose E(3)-transformations in an O(3)-transformation (rotation and/or reflection) followed by a translation. In this work we will mainly focus on dealing with O(3) transformations as translations are trivially dealt with. When representing the group elements of O(3) with matrices $\mathbf{R}$, as we have done before, the group product and inverse are simply given by the matrix-product and matrix-inverse. I.e., with $g = \mathbf{R}, g' = \mathbf{R}' \in G = \mathrm{O}(3)$ the group product and inverse are defined by

$$g \cdot g' := \mathbf{R}\mathbf{R}',$$
$$g^{-1} := \mathbf{R}^{-1}.$$

The group acts on $\mathbb{R}^3$ by matrix-vector multiplication, i.e., $g \cdot \mathbf{y} := \mathbf{R}\mathbf{y}$. The group elements of O(3) are square matrices with determinant $-1$ or $1$. Their action on $\mathbb{R}^3$ defines a reflection and/or rotation.

The special orthogonal group SO(3) has the same group product and inverse, but excludes reflections. The group thus consists of matrices with determinant $1$.

**The sphere $S^2$ is a homogeneous space of SO$(3)$.** The sphere is not a group as we cannot define a group product on $S^2$ that satisfies the group axioms. It can be convenient to treat it as a homogeneous space of the groups O(3) or SO(3). A space $\mathcal{X}$ is called a homogeneous space of a group $G$ if for any two points $x, y \in \mathcal{X}$ there exists a group element $g \in G$ such that $y = g \cdot x$.

The sphere $S^2$ is a homogeneous space of the rotation group SO(3) since any point on the sphere can be reached via the rotation of some reference vector. Consider for example an XYX parametrisation of SO(3) rotations in which three rotations are applied one after another via

$$\mathbf{R}_{\alpha,\beta,\gamma} = \mathbf{R}_{\alpha,\mathbf{n}_x}\mathbf{R}_{\beta,\mathbf{n}_y}\mathbf{R}_{\gamma,\mathbf{n}_x}, \tag{A.1}$$

with $\mathbf{n}_x$ and $\mathbf{n}_y$ denoting unit vectors along the $x$ and $y$ axis, and $\mathbf{R}_{\alpha,\mathbf{n}_x}$ denotes a rotation of $\alpha$ degrees around axis $\mathbf{n}_x$. We can model points on the sphere in a similar way via Euler angles via

$$\mathbf{n}_{\alpha,\beta} := \mathbf{R}_{\alpha,\beta,0}\mathbf{n}_x. \tag{A.2}$$

So, with two rotation angles, any point on the sphere can be reached. In the above we set $\gamma = 0$ in the parametrisation of the rotation matrix (an element from SO(3)) that rotates the reference vector $\mathbf{n}_x$, but it should be clear that with any $\gamma$ the same point $\mathbf{n}_{\alpha,\gamma}$ is reached. This means that there are many group elements in SO(3) that all map $\mathbf{n}_x$ to the same point on the sphere.

## A.2 INVARIANCE, EQUIVARIANCE AND REPRESENTATIONS

**Group representations.** We previously defined the group product, which tells us how elements of a group interact. We also showed that the groups E(3) and O(3) can transform the three dimensional vector space $\mathbb{R}^3$ via the group action. We usually think of the groups E(3) and O(3) as groups that describe transformations on $\mathbb{R}^3$, but these groups are not restricted to transformations on $\mathbb{R}^3$ and can generally act on arbitrary vector spaces via representations. A *representation* is an invertible linear

transformation $\rho(g) : V \to V$ parametrised by group elements $g \in G$ that acts on some vector space $V$, and which follows the group structure (it is a group homomorphism) via

$$\rho(g)\rho(h)v = \rho(g \cdot h)v \,,$$

with $v \in V$.

A representation can act on *infinite-dimensional vector spaces* such as functions. E.g., the so-called *left-regular representations* $\mathcal{L}_g$ of E(3) on functions $f : \mathbb{R}^3 \to \mathbb{R}$ on $\mathbb{R}^3$ is given by

$$\mathcal{L}_g[f](\mathbf{x}) = f(g^{-1}\mathbf{x}) \,,$$

i.e., it transforms the function by letting $g^{-1}$ act on the domain from the left. Here we used the notation $\mathcal{L}_g[f]$ to indicate that $\mathcal{L}_g$ transforms the function $f$ first, which creates a new function $\mathcal{L}(g)[f](\mathbf{x})$, which is then sampled at $\mathbf{x}$.

When representations transform *finite dimensional vectors* $v \in V = \mathbb{R}^d$, they are $d \times d$ dimensional matrices. In this work, we denote such *matrix representations* with boldface $\mathbf{D}$. A familiar example of a matrix representation of O(3) on $\mathbb{R}^3$ are the matrices $g = \mathbf{R} \in \mathrm{O}(3)$ themselves, i.e., $\mathbf{D}(g)\mathbf{x} = \mathbf{R}\mathbf{x}$.

Finally, any two representations, say $\mathbf{D}(g)$ and $\mathbf{D}'(g)$, are *equivalent* if they relate via a similarity transform via

$$\mathbf{D}'(g) = \mathbf{Q}^{-1}\mathbf{D}(g)\mathbf{Q} \,,$$

i.e., such representations describe one and the same thing but in a different basis, and the change of basis is carried out by $\mathbf{Q}$. Now that representations have been introduced we can formally define equivariance.

**Invariance and equivariance.** Equivariance is a property of an operator $\phi : \mathcal{X} \to \mathcal{Y}$ that maps between input and output vector spaces $\mathcal{X}$ and $\mathcal{Y}$. Given a group $G$ and its representations $\rho^{\mathcal{X}}$ and $\rho^{\mathcal{Y}}$ which transform vectors in $\mathcal{X}$ and $\mathcal{Y}$ respectively, an *operator $\phi : \mathcal{X} \to \mathcal{Y}$ is said to be equivariant if it satisfies the following constraint*

$$\rho^{\mathcal{Y}}(g)[\phi(x)] = \phi(\rho^{\mathcal{X}}(g)[x]) \,, \text{ for all } g \in G, x \in \mathcal{X} \,. \tag{A.3}$$

Thus, with an equivariant map, the output transforms predictably with transformations on the input. One might say that no information gets lost when the input is transformed, merely re-structured. One way to interpret Eq. (A.3) is therefore that the operators $\rho^{\mathcal{X}}(g) : \mathcal{X} \to \mathcal{X}$ and $\rho^{\mathcal{Y}}(g) : \mathcal{Y} \to \mathcal{Y}$ describe the same transformation, but in different spaces.

Invariance is a special case of equivariance in which $\rho^{\mathcal{Y}} = \mathcal{I}^{\mathcal{Y}}$ for all $g \in G$. I.e., *an operator $\phi : \mathcal{X} \to \mathcal{Y}$ is said to be invariant if it satisfies the following constraint*

$$\phi(x) = \phi(\rho^{\mathcal{X}}(g)[x]) \,, \text{ for all } g \in G, x \in \mathcal{X} \,. \tag{A.4}$$

Thus, with an invariant operator, the output of $\phi$ is unaffected by transformations applied to the input.

### A.3 Steerable vectors, Wigner-D matrices and irreducible representations

One strategy to build equivariant MLPs is to define input and output spaces of the MLPs and define how these spaces transform under the action of a group. This then sets an equivariance constraint on the operator that maps between these spaces. By only working with such equivariant operators we can guarantee that the entire learning framework is equivariant.

In our work, the proposed graph neural networks are translation equivariant by construction as any form of spatial information only enters the pipeline in the form of relative positions between nodes $(\mathbf{x}_j - \mathbf{x}_i)$. Then, any remaining operations are designed to be O(3) equivariant such that, together with the given translation equivariance, the complete framework is fully E(3) equivariant. Since translations are trivially dealt with, we focus on SO(3) and O(3) and show how to build equivariant MLPs through the use of the Clebsch-Gordan tensor product.

**Wigner-D matrices are irreducible representations.** For SO(3) there exists a collection of representations, indexed with their order $l \geq 0$, which act on vector spaces of dimension $2l + 1$. These representations are called Wigner-D matrices and we denote them with $\mathbf{D}^{(l)}(g)$. The use of Wigner-D

matrices is motivated by the fact that any matrix representation $\mathbf{D}(g)$ of SO(3) that acts on some vector space $V$ can be "reduced" to an equivalent block diagonal matrix representation with Wigner-D matrices along the diagonal:

$$\mathbf{D}(g) = \mathbf{Q}^{-1}(\mathbf{D}^{(l_1)}(g) \oplus \mathbf{D}^{(l_2)}(g) \oplus \dots)\mathbf{Q} = \mathbf{Q}^{-1} \begin{pmatrix} \mathbf{D}^{(l_1)}(g) & & \\ & \mathbf{D}^{(l_2)}(g) & \\ & & \ddots \end{pmatrix} \mathbf{Q}, \quad (A.5)$$

with $\mathbf{Q}$ the change of basis that makes them equivalent. The individual Wigner-D matrices themselves cannot be reduced and are hence *irreducible representations* of SO(3). Thus, since the block diagonal representations are equivalent to $\mathbf{D}$ we may as well work with them instead. This is convenient since each block, i.e., each Wigner-D matrix $\mathbf{D}^{(l_i)}$, only acts on a sub-space $V_{l_1}$ of $V$. As such we can factorise $V = V_{l_1} \oplus V_{l_2} \oplus \dots$, which motivates the use of steerable vector spaces and their direct sums as presented in Sec. 2.

The Wigner-D matrices are the irreducible representations of SO(3), but we can easily adapt these representations to be suitable for O(3) by including the group of reflections as a direct product. We will still refer to these representations as Wigner-D matrices in the entirety of this work, opting to avoid the distinction in favour of clarity of exposition. We further remark that explicit forms of the Wigner-D matrices can e.g. be found books such as in Sakurai & Napolitano (2017) and their numerical implementations in code libraries such as the e3nn library (Geiger et al., 2021a).

**Steerable vector spaces.** The $(2l + 1)$-dimensional vector space on which a Wigner-D matrix of order $l$ acts will be called *a type $l$ steerable vector space* and is denoted with $V_l$. E.g., a type-3 vector $\mathbf{h} \in V_3$ is transformed by $g \in$ O(3) via $\mathbf{h} \mapsto \mathbf{D}_3(g)\mathbf{h}$. We remark that this definition is equivalent to the definition of *steerable functions* commonly used in computer vision (Freeman et al., 1991; Hel-Or & Teo, 1996) via the viewpoint that steerable vectors can be regarded as the basis coefficients of a function expanded in a spherical harmonic basis. We elicit this viewpoint in Sec. A.4 and B.

At this point we are already familiar with type-0 and type-1 steerable vector spaces. Namely, type-0 vectors $h \in V_0 = \mathbb{R}$ are scalars, which are invariant to transformations $g \in$ O(3), i.e., $\mathbf{D}_0(g)h = h$. Type-1 features are vectors $\mathbf{h} \in \mathbb{R}^3$ which transform directly via the matrix representation of the group, i.e., $\mathbf{D}_1(g)\mathbf{h} = \mathbf{Rh}$.

### A.4 SPHERICAL HARMONICS

**Spherical harmonics.** Related to the Wigner-D matrices and their steerable vector spaces are the spherical harmonics[6]. Spherical harmonics are a class of functions on the sphere $S^2$, akin to the familiar circular harmonics that are best known as the 1D Fourier basis. As with a Fourier basis, spherical harmonics form an orthonormal basis for functions on $S^2$. In this work we use the real-valued spherical harmonics and denote them with $Y_m^{(l)} : S^2 \to \mathbb{R}$.

**Spherical Harmonics are Wigner-D functions.** One can also think of spherical harmonics as functions $\tilde{Y}_m^{(l)} : $ SO(3) $\to \mathbb{R}$ on SO(3) that are invariant to a sub-group of rotations via

$$Y_m^{(l)}(\mathbf{n}_{\alpha,\beta}) = Y_m^{(l)}(\mathbf{R}_{\alpha,\beta,\gamma}\mathbf{n}_x) =: \tilde{Y}_m^{(l)}(\mathbf{R}_{\alpha,\beta,\gamma}),$$

in which we used the parametrisation for $S^2$ and O(3) given in (A.2) and (A.1) respectively. Then, by definition, $\tilde{Y}_m^{(l)}$ is invariant with respect to rotation angle $\gamma$, i.e., $\forall_{\gamma \in [0,2\pi)} : \tilde{Y}_m^{(l)}(\mathbf{R}_{\alpha,\beta,\gamma}) = \tilde{Y}_m^{(l)}(\mathbf{R}_{\alpha,\beta,0})$. This viewpoint of regarding the spherical harmonics as $\gamma$-invariant functions on O(3) helps us to draw the connection to the Wigner-D functions $D_{mn}^{(l)}$ that make up the $2l + 1 \times 2l + 1$ elements of the Wigner-D matrices. Namely, the $n = 0$ column of Wigner-D functions are also $\gamma$-invariant and, in fact, correspond (up to a normalisation factor) to the spherical harmonics via

$$Y_m^{(l)}(\mathbf{n}_{\alpha,\beta}) = \frac{1}{\sqrt{2l + 1}} D_{m0}^{(l)}(\mathbf{R}_{\alpha,\beta,\gamma}). \quad (A.6)$$

---

[6]Solutions to Laplace's equation are called harmonics. Solutions of Laplace's equation on the sphere are therefore called spherical harmonics.

**The mapping from vectors into spherical harmonic coefficients is equivariant.** It then directly follows that vectors of spherical harmonics are steerable by the Wigner-D matrices of the same degree. Let $\mathbf{a}^{(l)}(\mathbf{n}) := (Y_{-l}^{(l)}(\mathbf{n}), \ldots, Y_l^{(l)}(\mathbf{n}))^T$ be the embedding of a direction vector $\mathbf{n} \in S^2$ in spherical harmonics. Then this vector embedding is equivariant as it satisfies

$$\forall_{\mathbf{R}' \in \mathrm{SO}(3)} \ \forall_{\mathbf{n} \in S^2} : \quad \mathbf{a}^{(l)}(\mathbf{R}'\mathbf{n}) = \mathbf{D}^{(l)}(\mathbf{R}')\mathbf{a}^{(l)}(\mathbf{n}) . \tag{A.7}$$

Using the $S^2$ and O(3) parametrization of (A.2) and (A.1) this is derived as follows

$$\mathbf{a}^{(l)}(\mathbf{R}'\mathbf{n}_{\alpha,\beta}) = \begin{pmatrix} Y_{-l}^{(l)}(\mathbf{R}'\mathbf{n}_{\alpha,\beta}) \\ \vdots \\ Y_l^{(l)}(\mathbf{R}'\mathbf{n}_{\alpha,\beta}) \end{pmatrix} = \begin{pmatrix} D_{-l0}^{(l)}(\mathbf{R}'\mathbf{R}_{\alpha,\beta,\gamma}) \\ \vdots \\ D_{l0}^{(l)}(\mathbf{R}'\mathbf{R}_{\alpha,\beta,\gamma}) \end{pmatrix} = \mathbf{D}^{(l)}(\mathbf{R}') \begin{pmatrix} D_{-l0}^{(l)}(\mathbf{R}_{\alpha,\beta,\gamma}) \\ \vdots \\ D_{l0}^{(l)}(\mathbf{R}_{\alpha,\beta,\gamma}) \end{pmatrix}$$

$$= \mathbf{D}^{(l)}(\mathbf{R}') \begin{pmatrix} Y_{-l}^{(l)}(\mathbf{n}_{\alpha,\beta}) \\ \vdots \\ Y_l^{(l)}(\mathbf{n}_{\alpha,\beta}) \end{pmatrix} = \mathbf{D}^{(l)}(\mathbf{R}')\mathbf{a}^{(l)}(\mathbf{n}_{\alpha,\beta}) .$$

**Steerable vectors represent steerable functions on $S^2$.** Just like the 1D Fourier basis forms a complete orthonormal basis for 1D functions, the spherical harmonics $Y_m^{(l)}$ form an orthonormal basis for $\mathbb{L}_2(S^2)$, the space of square integrable functions on the sphere. Any function on the sphere can thus be represented by a steerable vector $\tilde{\mathbf{a}} \in V_0 \oplus V_1 \oplus \ldots$ when it is expressed in a spherical harmonic basis via

$$f(\mathbf{n}) = \sum_{l \geq 0} \sum_{m=-l}^{l} a_m^{(l)} Y_m^{(l)}(\mathbf{n}) . \tag{A.8}$$

Since spherical harmonics form an orthonormal basis, the coefficient vector $\mathbf{a}$ can directly be obtained by taking $\mathbb{L}_2(S^2)$-inner products of the function $f$ with the spherical harmonics, i.e. ,

$$a_m^{(l)} = (f, Y_m^{(l)})_{\mathbb{L}_2(S^2)} = \int_{S^2} f(\mathbf{n}) Y_m^{(l)}(\mathbf{n}) \mathrm{d}\mathbf{n} . \tag{A.9}$$

Equation (A.9) is sometimes referred to as the Fourier transform on $S^2$, and Eq. (A.8) as the inverse spherical Fourier transform. Thus, one can identify steerable vectors $\mathbf{a}$ with functions on the sphere $S^2$ via the spherical Fourier transform.

In this paper, we visualise such functions on $S^2$ via glyph-visualisations which are obtained as surface plots by taking $\mathbf{n} \in S^2$ and scaling it by $|f(\mathbf{n})| = \mathrm{sign}(f(\mathbf{n})) f(\mathbf{n})$:

$$\{ \mathbf{n}|f(\mathbf{n})| \ \mid \ \mathbf{n} \in S^2 \} \quad \Longleftrightarrow \quad$$

where each point on this surface is color-coded with the function value $f(\mathbf{n})$. The visualisations are thus color-coded spheres that are stretched in each direction $\mathbf{n}$ via $\|f(\mathbf{n})\|$.

**Steerable vectors also represent steerable functions on O(3).** In order to draw a connection between group equivariant message passing and group convolutions, as we did in Sec. 3 of the main paper, it is important to understand that steerable vectors also represent functions on the group SO(3) via an SO(3)-Fourier transform. The collection of $(2l + 1)^2$ Wigner-D functions $D_{mn}^{(l)}$ form an orthonormal basis for $\mathbb{L}_2(\mathrm{SO}(3))$. This orthonormal basis allows for a Fourier transform that maps between the function space $\mathbb{L}_2(\mathrm{SO}(3))$ and steerable vector space $V$; the forward and inverse Fourier transform on SO(3) are respectively given by

$$a_{mn}^{(l)} = \int_{\mathrm{O}(3)} f(g) D_{mn}^{(l)}(g) \mathrm{d}g , \tag{A.10}$$

$$f(g) = \sum_{l \geq 0} \sum_{m=-l}^{l} \sum_{n=-l}^{l} a_{mn}^{(l)} D_{mn}^{(l)}(g) , \tag{A.11}$$

with $\mathrm{d}g$ the Haar measure of the group. Noteworthy, the forward Fourier transform generates a matrix of Fourier coefficients, rather than a vector in spherical case. The coeffecient matrix is steerable by left-multiplication with the Wigner-D matrices of the same type $\mathbf{D}^{(l)}$.

## A.5 CLEBSCH-GORDAN PRODUCT AND STEERABLE MLPS

In a regular MLP one maps between input and output vector spaces linearly via matrix-vector multiplication and applies non-linearities afterwards. In steerable MLPs, one maps between steerable input and output vector spaces via the Clebsch-Gordan tensor product and applies non-linearities afterwards. Akin to the learnable weight matrix in regular MLPs, the learnable Glebsch-Gordan tensor product is the main workhorse of our steerable MLPs.

**Clebsch-Gordan tensor product.** The Clebsch-Gordan (CG) tensor product allows us to map between steerable input and output spaces. While there is much to be said about tensors and tensor products in general, we here intend to focus on intuition. In general, a tensor product involves the multiplication between all components of two input vectors. E.g., with two vectors $\mathbf{h}_1 \in \mathbb{R}^{d_1}$ and $\mathbf{h}_2 \in \mathbb{R}^{d_2}$, the tensor product is given by

$$\mathbf{h}_1 \otimes \mathbf{h}_2 = \mathbf{h}_1 \mathbf{h}_2^T = \begin{pmatrix} h_1 h_1 & h_1 h_2 & \dots \\ h_2 h_1 & h_2 h_2 & \dots \\ \vdots & \vdots & \ddots \end{pmatrix} ,$$

which we can flatten into a $d_1 d_2$-dimensional vector via an operation which we denote with $\mathrm{vec}(\mathbf{h}_1 \otimes \mathbf{h}_2)$. In our steerable setting we would like to work exclusively with steerable vectors and as such we would like for any two steerable vectors $\tilde{\mathbf{h}}_1 \in V_{l_1}$ and $\tilde{\mathbf{h}}_2 \in V_{l_2}$, that the tensor product's output is again steerable with a O(3) representation $\mathbf{D}(g)$ such that the following equivariance constraint is satisfied:

$$\mathbf{D}(g)(\tilde{\mathbf{h}}_1 \otimes \tilde{\mathbf{h}}_2) = (\mathbf{D}^{(l_1)}(g)\tilde{\mathbf{h}}_1) \otimes (\mathbf{D}^{(l_2)}(g)\tilde{\mathbf{h}}_2) . \tag{A.12}$$

Via the identity $\mathrm{vec}(\mathbf{A}\mathbf{X}\mathbf{B}) = (\mathbf{B}^T \otimes \mathbf{A})\mathrm{vec}(\mathbf{X})$, we can show that the output is indeed steerable:

$$\mathrm{vec}\left( (\mathbf{D}^{(l_1)}(g)\tilde{\mathbf{h}}_1)(\mathbf{D}^{(l_2)}(g)\tilde{\mathbf{h}}_2)^T \right) = \mathrm{vec}\left( \mathbf{D}^{(l_1)}(g)\tilde{\mathbf{h}}_1 \tilde{\mathbf{h}}_2^T \mathbf{D}^{(l_2)T}(g) \right)$$
$$= \left( \mathbf{D}^{(l_2)}(g) \otimes \mathbf{D}^{(l_1)}(g) \right) \mathrm{vec}\left( \tilde{\mathbf{h}}_1 \tilde{\mathbf{h}}_2^T \right) .$$

The resulting vector $\mathrm{vec}(\tilde{\mathbf{h}}_1 \otimes \tilde{\mathbf{h}}_2)$ is thus steered by a representation $\mathbf{D}(g) = \mathbf{D}^{(l_2)}(g) \otimes \mathbf{D}^{(l_1)}(g)$. Since any matrix representation of O(3) can be reduced to a direct sum of Wigner-D matrices (see (A.5)), the resulting vector can be organised via a change of basis into parts that individually transform via Wigner-D matrices of different type. I.e. $\tilde{\mathbf{h}} = \mathrm{vec}(\tilde{\mathbf{h}}_1 \otimes \tilde{\mathbf{h}}_2) \in V = V_0 \oplus V_1 \oplus \dots$, with $V_l$ the steerable sub-vector spaces of type $l$.

With the CG tensor product we directly obtain the vector components for the steerable sub-vectors of type $l$ as follows. Let $\tilde{\mathbf{h}}^{(l)} \in V_l = \mathbb{R}^{2l+1}$ denote a steerable vector of type $l$ and $h_m^{(l)}$ its components with $m = -l, -l+1, \dots, l$. Then the $m$-th component of the type $l$ sub-vector of the output of the tensor product between two steerable vectors of type $l_1$ and $l_2$ is given by

$$(\tilde{\mathbf{h}}^{(l_1)} \otimes_{cg} \tilde{\mathbf{h}}^{(l_2)})_m^{(l)} = w \sum_{m_1=-l_1}^{l_1} \sum_{m_2=-l_2}^{l_2} C_{(l_1,m_1)(l_2,m_2)}^{(l,m)} h_{m_1}^{(l_1)} h_{m_2}^{(l_2)} , \tag{A.13}$$

in which $w$ is a learnable parameter that scales the product and $C_{(l_1,m_1)(l_2,m_2)}^{(l,m)}$ are the Clebsch-Gordan coefficients. The CG tensor product is a sparse tensor product, as generally many of the $C_{(l_1,m_1)(l_2,m_2)}^{(l,m)}$ components are zero. Most notably, $C_{(l_1,m_1)(l_2,m_2)}^{(l,m)} = 0$ whenever $l < |l_1 - l_2|$ or $l > l_1 + l_2$. E.g., a type-0 and a type-1 feature vector cannot create a type-2 feature vector. Well known examples of the CG product are the scalar product ($l_1 = 0, l_2 = 1, l = 1$), which takes as input a scalar and a type-1 vector to generate a type-1 vector, the dot product ($l_1 = 1, l_2 = 1, l = 0$) and cross product ($l_1 = 1, l_2 = 1, l = 1$).

Examples of instances of the CG product are:

- The product of two scalars is a CG product which takes as input two type-0 "vectors" to generate another scalar ($l_1 = 0, l_2 = 0, l = 0$).

- The scalar product, which takes as input a scalar and a type-1 vector to generate a type-1 vector ($l_1 = 0, l_2 = 1, l = 1$).

- The dot product, which takes two type-1 vectors as input to produce a scalar ($l_1 = 1, l_2 = 1, l = 0$).

- The cross product, which takes two type-1 vectors as input to produce another type-1 vector ($l_1 = 1, l_2 = 1, l = 1$).

In a standard linear layer, an input vector is transformed via multiplication with the weight matrix, where weights and elements of the input vector are simply multiplied. For $l > 0$ elements of the Clebsch-Gordan tensor product of Eq. (A.13) more than one mathematical operation is needed for the connection with one weight, as can easily be verified if $h_{m_1}^{(l_1)}$ and $h_{m_2}^{(l_2)}$ are thought of as e.g. type-1 vectors. This makes calculation of the Clebsch-Gordan tensor product slow for higher order irreps, as can be observed in all experiments.

**Steerable MLPs.** The CG tensor product thus allows us to map two steerable input vectors to a new output vector and can furthermore be extended to define a tensor product between steerable vectors of mixed types. The CG tensor product can be parametrised by weights where the product is scaled with some weight $w$ for each triplet of types $(l_1, l_2, l)$ for which the CG coefficients are non-zero[7]. We indicate such CG tensor products with $\otimes_{cg}^{\mathbf{W}}$. While in principle the CG tensor product takes two steerable vectors as input, in this work we mainly use it with one of its input vectors "fixed". The CG tensor product can then be regarded as a linear layer that maps between two steerable vector spaces and we denote this $\mathbf{W}_{\tilde{\mathbf{a}}}\tilde{\mathbf{h}} := \tilde{\mathbf{h}} \otimes_{cg}^{\mathbf{W}} \tilde{\mathbf{a}}$, with $\tilde{\mathbf{a}}$ the steerable vector that is considered to be fixed. With this viewpoint we can design MLPs in the same way as we are used to with regular linear layers, and establish clear analogies with convolution layers (Sec. 3 of the main paper).

While in general the CG tensor product between two steerable vectors of type $l_1$ and $l_2$ can contain steerable vectors up to degree $l = l_1 + l_2$, one typically "band-limits" the output vector by only considering the steerable vectors up to some degree $l_{max}$. In our experiments we band-limit the hidden feature representations to $l_{max} = l_f$.

Finally, the amount of interaction between the steerable sub-vectors in the hidden representations of degree $\tilde{\mathbf{h}} \in V_{L=l_f} = V_0 \oplus V_1 \oplus \ldots V_{l_f}$ is determined by the maximum order $l_a$ of the steerable vector $\tilde{\mathbf{a}}$ on which the steerable linear layer is conditioned. Namely, the CG tensor product only produces type $l$ steerable sub-vectors for $|l_f - l_a| \geq l \leq |l_f + l_a|$. For example, it is not sensible to embed positions as steerable vectors up to order $l_a = 5$ when the hidden representations are of degree $l_f = 2$; the lowest steerable vector type that can be produced with the CG product of a $l = 5$ sub-vector of $\tilde{\mathbf{a}}$ with any sub-vector of the hidden representation vector will be $l = 3$ and since the hidden representations are band-limited to a maximum type of $l_f = 2$ higher order vectors will be ignored.

## B  STEERABLE GROUP CONVOLUTIONS

**Steerable functions.** Through the equivalence of steerable vectors and spherical functions via the spherical Fourier transform, it is clear that our definition of *steerable vectors* coincides with the more common definition of *steerable functions*, as commonly used in computer vision (Freeman et al., 1991). A function $f$ is called steerable (Hel-Or & Teo, 1996) under a transformation group $G$ if any transformation $g \in G$ on $f$ can be written as a linear combination of a fixed, finite set of $n$ basis functions $\{\phi_i\}$:

$$\mathcal{L}_g f = \sum_{i=1}^{n} \alpha_i(g)\phi_i = \boldsymbol{\alpha}^T(g)\boldsymbol{\Phi} \,, \tag{B.1}$$

in which $\mathcal{L}_g$ is the left regular representation of the group $G$ that performs the transformation on $f$ (see section A.2). In case of functions in a spherical harmonic basis, which we denote with

---

[7] In terms of the `e3nn` library (Geiger et al., 2021a) one then says a path exists between these 3 types.

$f_{\mathbf{a}} : \sum_{l \geq 0} \sum_{m=-l}^{l} a_m^{(l)} Y_m^l(\mathbf{n})$, it directly follows from Eq. (A.7) that they are steerable via

$$\mathcal{L}_g f_{\mathbf{a}}(\mathbf{n}) = f_{\mathbf{D}(g)\mathbf{a}}(\mathbf{n}) \,. \tag{B.2}$$

In terms of the steerable function definition in Eq. (B.1), this means that $\boldsymbol{\alpha}(g) = \mathbf{D}(g)\mathbf{a}$ and $\boldsymbol{\Phi} = (Y_0^{(0)}, Y_{-1}^{(1)}, Y_0^{(1)}, \dots)^T$, i.e. the set of basis functions flattened into a vector.

**A translation-rotation template machting motivation for (steerable) group convolutions.**    The notion of steerability becomes particularly clear when viewed in the computer vision context (Freeman et al., 1991), where one may be interested in detecting visual features under arbitrary rotations. Let us consider the case of 3D cross-correlations of a kernel $k : \mathbb{R}^3 \to \mathbb{R}$ with an input feature map $f : \mathbb{R}^3 \to \mathbb{R}$:

$$f'(\mathbf{x}) = (k \star f)(\mathbf{x}) = \int_{\mathbb{R}^3} k(\mathbf{x}' - \mathbf{x}) f(\mathbf{x}') \mathrm{d}\mathbf{x}' \,. \tag{B.3}$$

We think of such a kernel $k$ as describing a particular visual pattern. In many applications, we want to detect such patterns under arbitrary rotations. For example, in 3D medical image data there is no preferred orientation and features (such as blood vessels, lesions, ...) can appear under any angle, and the same holds for particular atomic patterns in molecules. So ideally, one wants to apply the convolution kernel under all such transformations and obtain feature maps via

$$f'(\mathbf{x}, \mathbf{R}) = \int_{\mathbb{R}^3} k(\mathbf{R}^{-1}(\mathbf{x}' - \mathbf{x})) f(\mathbf{x}') \mathrm{d}\mathbf{x}' \,. \tag{B.4}$$

By repeating the convolutions with rotated kernels we are able to detect the presence of a certain feature at all possible angles. In a group convolution context (Cohen & Welling, 2016), the above is what one usually calls a lifting group convolution (Bekkers, 2019) (feature maps are lifted from $\mathbb{R}^3$ to the group SE(3)).

**Group convolutions.**    In *regular group convolutional neural networks* one continues to work with such higher dimensional feature maps in which the kernels are also functions on the group. The lifting and subsequent group convolutions then all have the same form and are defined via the group action on $\mathbb{R}^3$ and group product respectively via

$$f'(g) = \int_{\mathbb{R}^3} k(g^{-1} \cdot \mathbf{x}') f(\mathbf{x}') \mathrm{d}\mathbf{x}' \,, \tag{B.5}$$

$$f'(g) = \int_{SE(3)} k(g^{-1} \cdot g') f(g') \mathrm{d}g' \,, \tag{B.6}$$

where $g \in$ SE(3), $\mathrm{d}g$ the Haar measure on the group and where $\cdot$ in (B.5) and (B.6) respectively denote the group action on $\mathbb{R}^3$ and group product of SE(3) (cf. Section A.1). Note that Eq. (B.5) is exactly the same as Eq. (B.4) but in different notation.

The lifting group convolution thus generates a function on the joint space of positions $\mathbb{R}^3$ and rotations SO(3). In numerical implementations, this space needs to be discretised, i.e., for a particular finite grid of rotations we want to store the results of the convolutions for each rotation $\mathbf{R}$. This approach then requires that the convolution kernel is continuous and can be sampled under all transformations. Hence, such kernels can be expanded in a continuous basis such as spherical harmonics (Weiler et al., 2018), B-splines (Bekkers, 2019) or they can be parametrised via MLPs (Finzi et al., 2020). Alternatively, the kernels are only transformed via a sub-group of transformations in E(3) that leaves the grid on which the kernel is defined intact, as in (Worrall & Brostow, 2018; Winkels & Cohen, 2018). An advantage of regular group convolution methods is that normal point-wise activation functions can directly be applied to the feature maps; a down-side is that these methods are only equivariant to the sub-group on which they are discretised. When expressing the convolution kernel in terms of spherical harmonics, however, there is no need for such a discretisation at all and one can obtain the response of the convolution at any rotation $\mathbf{R}$ after convolving with the basis functions. This works as follows.

**Steerable template matching.**    Suppose a 3D convolution kernel that is expanded in a spherical harmonic basis up to degree $L$ as follows

$$k(\mathbf{x}) = k_{\tilde{\mathbf{c}}(\|\mathbf{x}\|)}(\mathbf{x}) := \sum_{l}^{L} \sum_{m=-l}^{l} c_m^{(l)}(\|\mathbf{x}\|) Y_m^{(l)} \left( \tfrac{\mathbf{x}}{\|\mathbf{x}\|} \right) \,, \tag{B.7}$$

with $\tilde{\mathbf{c}} = (c_0^{(0)}, c_{-1}^{(1)}, c_0^{(1)} \dots)^T$ the vector of basis coefficients that can depend on $\|\mathbf{x}\|$. The coefficients can e.g. be parametrised with an MLP that takes as input $\|\mathbf{x}\|$ and returns the coefficient vector. We note that such coefficients are then O(3) invariant, i.e., $\forall_{\mathbf{R} \in O(3)} : \tilde{\mathbf{c}}(\|\mathbf{Rx}\|) = \tilde{\mathbf{c}}(\|\mathbf{x}\|)$. Furthermore, we labelled the vector with a "˜" to indicate it is a steerable vector as it represents the coefficients relative to a spherical harmonic basis. It then follows (from Eq. (B.2)) that the kernel is steerable via

$$k(\mathbf{R}^{-1}\mathbf{x}) = k_{\mathbf{D}(\mathbf{R})\tilde{\mathbf{c}}(\|\mathbf{x}\|)}(\mathbf{x}) \,,$$

i.e., via a transformation of the coefficient vector $\tilde{\mathbf{c}}$ by it O(3) representation $\mathbf{D}(\mathbf{R})$.

This steerability property, together with linearity of the convolutions and basis expansion, implies that with such steerable convolution kernels we can obtain their convolutional response at any rotation directly from convolutions with the basis functions. Instead of first expanding the kernel in the basis by taking a weighted sum of basis functions with their corresponding coefficients, and only then doing the convolutions, we can change this order and first do the convolution with the basis functions and sum afterwards. In doing so we create a vector of responses $\tilde{\mathbf{f}}(\mathbf{x}) = (f_0^{(0)}(\mathbf{x}), f_{-1}^{(1)}(\mathbf{x}), f_0^{(1)}(\mathbf{x}), \dots)^T$ of which the elements are given by

$$f_m^{(l)}(\mathbf{x}) = ((c_m^l Y_m^{(l)}) \star f^{in})(\mathbf{x}) \tag{B.8}$$
$$= \int_{\mathbb{R}^3} c_m^{(l)}(\|\mathbf{x}\|) Y_m^{(l)}(\mathbf{x}' - \mathbf{x}) f(\mathbf{x}') \mathrm{d}\mathbf{x}' \,.$$

Then the result with the convolution kernel (Eq. (B.7)) is obtained simply by a sum over the vector components which we denote with $\mathrm{SumReduce}_{l,m}$ as follows

$$f'(\mathbf{x}) = \mathrm{SumReduce}_{l,m}(\tilde{\mathbf{f}}'(\mathbf{x})) := \sum_l^L \sum_{m=-l}^l f_m'^{(l)}(\mathbf{x}) \,.$$

If one were to be interested in the rotated filter response at $\mathbf{x}$ one can first rotate the steerable vector $\tilde{\mathbf{f}}'(\mathbf{x})$ via the matrix representation $\mathbf{D}(\mathbf{R})$ and only then do the reduction. I.e., once the convolutions of (B.8) are done the lifting group convolution result is directly obtained via

$$f'(\mathbf{x}, \mathbf{R}) = \mathrm{SumReduce}_{l,m}(\mathbf{D}(\mathbf{R})\tilde{\mathbf{f}}'(\mathbf{x})) \,. \tag{B.9}$$

A convolution with a kernel expressed in a spherical harmonics basis thus generates a signal on O(3) at each point $\mathbf{x}$.

When only considering the $m = 0$ component for each order $l$ in the kernel parametrisation, the kernels have an axial symmetry in them due to which the convolution results in a point-wise spherical signal. This is in fact the situation that we consider in our weighted CG products (Eq. 5) where we do not index the weights with $m$. This choice leads to computational efficiency. It is however not necessary to constrain the kernels as such and continue with steerable group convolutions on the full space $\mathbb{R}^3 \rtimes SO(3)$, this would require working with CG products with weights indexed by both $m$ and $m'$ (Weiler et al., 2018). Since in this work we choose to with steerable vector spaces $V = m_0 V_0 \oplus m_1 V_1 \oplus \dots$ with arbitrary multiplicities $m_l$ for each type, we implement what is called generalised steerable convolutions, of which the SO(3) convolutions are a special case with $m_l = (2l + 1)$ (Kondor et al., 2018).

**Steerable group convolutions.** When working with steerable convolution kernels one does not have to work with a grid on O(3). There are reasons to avoid working with a grid on O(3) as one cannot numerically obtain exact equivariance if the chosen grid is not a sub-group of O(3). When limiting to a discrete subgroups one can guarantee exact equivariance to the sub-group, but ideally one obtains equivariance to the entire group O(3). Steerable methods provide a way to build neural networks entirely independent of any sampling on O(3), since, as we have seen, steerable convolutions directly result in functions on the entire group O(3) via steerable vectors. That is, *at each location $\mathbf{x}$ we have a steerable vector $\tilde{\mathbf{f}}(\mathbf{x})$ which represents a function on O(3) via the inverse Fourier transform given in Eq. (A.11)*. In fact, the sum reduction in Eq. (B.9) corresponds to a discrete inverse Fourier transform.

Finally, let us revisit steerable group convolutions one more time but now in the steerable vector notation used throughout this paper. The steerable convolution, as described in (B.8), is an operator

that maps between steerable vector fields of different types. In the above example the input feature map was one that only provided a single scalar value per location $\mathbf{x}$, i.e. a type-0 steerable vector, and the output was a steerable vector field containing steerable vectors up to type $L$. The transition from type-0 vector field to a type-$L$ vector field happened via tensor products with steerable vectors of spherical harmonics, and this tensor product was parametrised by the coefficients $\tilde{c}$. Suppose a convolution of a single type-$l_1$ steerable input field with a convolution kernel that is expanded in a spherical harmonic basis of only type $l_2$ via $k(\mathbf{x}) = \sum_{m=-l_2}^{l_2} w(\|\mathbf{x}\|) Y_m^{(l)}\left(\frac{\mathbf{x}}{\|\mathbf{x}\|}\right)$. The kernel can then be represented as a steerable vector field in itself as $\tilde{\mathbf{k}}(\mathbf{x}) = w(\|\mathbf{x}\|)\tilde{\mathbf{a}}(\mathbf{x})$, with $\tilde{\mathbf{a}}$ the type-$l_2$ spherical harmonic embedding given in Eq. 5 of the main paper. Such a steerable convolution maps from input feature maps $\tilde{\mathbf{f}} : \mathbb{R}^3 \rightarrow V_{l_1}$ using a convolution kernel $\tilde{\mathbf{k}} : \mathbb{R}^3 \rightarrow V_{l_2}$ to an output $\tilde{\mathbf{f}}' : \mathbb{R}^3 \rightarrow V_l$ via

$$\tilde{\mathbf{f}}'(\mathbf{x}) = \int_{\mathbb{R}^3} \tilde{\mathbf{f}}(\mathbf{x}) \otimes_{cg}^{w(\|\mathbf{x}\|)} \tilde{\mathbf{a}}(\mathbf{x}' - \mathbf{x}) \mathrm{d}\mathbf{x}' \ .$$

Here we assumed steerable feature vector fields of a single type, but in general such convolutions can map between vector fields of mixed type analogous to the standard convolutional case in CNNs where mappings occur between multi-channel type-0 vector fields.

**Steerable and regular group convolutions.** In conclusion, both steerable and regular group convolutions produce feature maps on the entire group E(3) and they are equivalent when the regular convolution kernel is expanded in a steerable basis. In regular group convolutions, the response is stored on a particular grid which is e.g. the Cartesian product of a regular 3D grid with a particular discretisation of O(3). In regular group convolutions, we directly index the responses with $(\mathbf{x}, \mathbf{R}) \in \mathrm{E}(3) \mapsto \mathbf{f}(\mathbf{x}, \mathbf{R})$. In steerable convolutions the filter responses $\mathbf{x} \rightarrow \tilde{\mathbf{f}}(\mathbf{x})$ are stored at each spatial grid point $\mathbf{x}$ in steerable vectors $\tilde{\mathbf{f}}(\mathbf{x})$ from which functions on the full group O(3) can be obtained via an inverse Fourier transform[8]. As such one recovers the regular representation via $(\mathbf{x}, \mathbf{R}) \mapsto \mathcal{F}^{-1}[\tilde{\mathbf{f}}(\mathbf{x})](\mathbf{R})$. Regular group convolutions can however not be perfectly equivariant to all transformations in O(3) due to discretisaton artifacts, or they are only equivariant to a discrete sub-group of O(3). Steerable group convolutions on the other hand are exactly equivariant to all transformations in O(3) via steerable representations of O(3).

---

[8]This Fourier transform in fact enables working with classic point-wise activation functions that could be applied point-wise to each location in a spherical or O(3) signal as in (Cohen et al., 2018). It is important to note that one cannot readily apply activation functions such as ReLU directly to the steerable coefficients, but one could sample the O(3) signal on a grid via the inverse Fourier transform, apply the activation function, and transform back into the steerable basis. This is the approach taken in (Cohen et al., 2018). In this paper we work entirely in the steerable domain (O(3) Fourier space) and work with gated non-linearities (Weiler et al., 2018).

## C  Experiments

### C.1  Pseudocode of SEGNN and ablated architectures

In this subsection, we provide pseudocode for our implementation SEGNN and steerable ablations. Pseudocode for steerable equivariant point conv layers such as (Thomas et al., 2018) with messages as in Eq. (14) and regular gated non-linearities as activation/update function, labelled SE$_{\text{linear}}$, is outlined in Alg. 1. Pseudocode for the same network but with messages obtained in a non-linear manner via 2-layer steerable MLPs as in Eq. (16) is outlined in Alg. 2. This layer is labelled as SE$_{\text{non-linear}}$. Finally, pseudocode for SEGNN implementation is provided in Alg. 3. SEGNN layers allow for the inclusion of geometric and physical quantities via node attributes $\tilde{\mathbf{a}}_i$. These new node update layers can be regarded as new steerable activation function and allow a functionality which is not possible in layers such as SE$_{\text{linear}}$ or SE$_{\text{non-linear}}$. A more detailed explanation of the Clebsch-Gordan tensor product (CGTensorProduct), the spherical harmonic embedding (SphericalHarmonicEmbedding) and the application of gated non-linearities can be found in the `e3nn` (Geiger et al., 2021a) library.

**Injecting geometric and physical quantities.**  In the following algorithms the relative position vector $\mathbf{x}_{ij}$ between node $\mathbf{f}_i$ and node $\mathbf{f}_j$ is given as an example for a geometric quantity used to steer the message layers. To get more concrete, we look at Alg. 3 (SEGNN) and discuss edge attributes and node attributes. Steerable attributes first of all have to be embedded via spherical harmonics. Steerable edge attribute $\tilde{\mathbf{a}}_{ij}$ in most cases consist of relative positions. Instead of or additional to relative positions also relative forces or relative velocities could be used. And finally, also node specific quantities such as force can be added together to serve as edge attributes. Steerable node attributes are in most cases real physical quantities. In Alg. 3 they are sketched with $\mathbf{v}_i^1$ and $\mathbf{v}_i^2$, and might comprise velocity, acceleration, spin, or force. However, also adding the spherical harmonics embedding of relative positions at node $\mathbf{f}_i$ results in a steerable node attribute. Lastly, it is to note that if multiple steerable attributes exist they can be either added or concatenated. However, the latter results in significantly more weights in the CG tensor product and does not result in better performance in any of the conducted experiments.

**Gated non-linearities.**  The gate activation is a direct sum of two sets of irreps. The first set is the set of scalar irreps passed through activation functions. The second set is the set of higher order irreps, which is multiplied by an additional set of scalar irreps that is introduced solely for the activation layer and passed through activation functions. For example, in Alg. 1, the scalar irreps $\mathbf{g}_{ij}$ are the additional scalar irreps, introduced to "gate" the non-scalar irreps of $\tilde{\mathbf{m}}_{ij}$.

---

**Algorithm 1** Code sketch of an SE$_{\text{linear}}$ message passing layer. The SE$_{\text{linear}}$ layer updates steerable node features $\tilde{\mathbf{f}}_i' \leftarrow \tilde{\mathbf{f}}_i$. The additional scalar irreps $\mathbf{g}_i$ are used to "gate" the non-zero order irreps as introduced in Weiler et al. (2018). A fully documented code for this layer will be applicable in our repo.

---

**Require:** $\tilde{\mathbf{f}}_i, \mathbf{x}_{ij}$      ▷ Steerable nodes $\tilde{\mathbf{f}}_i$, relative position vector $\mathbf{x}_{ij}$ between node $\tilde{\mathbf{f}}_i$ and node $\tilde{\mathbf{f}}_j$

    **function** O3_Tensor_Product(input1, input2)
        output ← CGTensorProduct(input1, input2)      ▷ Apply CG tensor product following Eq. (6)
        output ← output + bias      ▷ Add bias to zero order irreps
        return output
    **end function**
    $\tilde{\mathbf{a}}_{ij} \leftarrow$ SphericalHarmonicEmbedding($\mathbf{x}_{ij}$)      ▷ Spherical harmonic embedding of $\mathbf{x}_{ij}$ (Eq. (4))
    $\tilde{\mathbf{h}}_{ij} \leftarrow \tilde{\mathbf{f}}_i \oplus \tilde{\mathbf{f}}_j \oplus \|\mathbf{x}_{ij}\|^2$      ▷ Concatenate input for messages between $\tilde{\mathbf{f}}_i, \tilde{\mathbf{f}}_j$
    $\tilde{\mathbf{m}}_{ij} \oplus \mathbf{g}_{ij} \leftarrow$ O3_Tensor_Product($\tilde{\mathbf{h}}_{ij}, \tilde{\mathbf{a}}_{ij}$)      ▷ Calculate messages $\tilde{\mathbf{m}}_{ij}$ and scalar irreps $\mathbf{g}_{ij}$
    $\tilde{\mathbf{m}}_i \oplus \mathbf{g}_i \leftarrow \sum_j \tilde{\mathbf{m}}_{ij} \oplus \sum_j \mathbf{g}_{ij}$      ▷ Aggregate messages $\tilde{\mathbf{m}}_{ij}$ and scalar irreps $\mathbf{g}_{ij}$
    $\tilde{\mathbf{f}}_i' \leftarrow \tilde{\mathbf{f}}_i + \text{Gate}(\tilde{\mathbf{m}}_i, \text{Swish}(\mathbf{g}_i))$      ▷ Transform nodes via gated non-linearities

---

**Algorithm 2** Code sketch of an $SE_{\text{non-linear}}$ message passing layer. The $SE_{\text{non-linear}}$ layer updates steerable node features $\tilde{\mathbf{f}}_i' \leftarrow \tilde{\mathbf{f}}_i$. The additional scalar irreps $\mathbf{g}_i$ are used to "gate" the non-zero order irreps as introduced in Weiler et al. (2018). A fully documented code for this layer will be applicable in our repo.

---

**Require:** $\tilde{\mathbf{f}}_i, \mathbf{x}_{ij}$      ▷ Steerable nodes $\tilde{\mathbf{f}}_i$, relative position vector $\mathbf{x}_{ij}$ between node $\tilde{\mathbf{f}}_i$ and node $\tilde{\mathbf{f}}_j$

  **function** O3_TENSOR_PRODUCT(input1, input2)
      output $\leftarrow$ CGTensorProduct(input1, input2)      ▷ Apply CG tensor product following Eq. (6)
      output $\leftarrow$ output + bias             ▷ Add bias to zero order irreps
      return output
  **end function**
  **function** O3_TENSOR_PRODUCT_SWISH_GATE(input1, input2)
      output $\oplus \mathbf{g}_i \leftarrow$ O3_TENSOR_PRODUCT(input1, input2)      ▷ Output plus scalar irreps $\mathbf{g}_i$
      output$_{\text{gated}} \leftarrow$ Gate(output, Swish($\mathbf{g}_i$))      ▷ Transform output via gated non-linearities
      return output
  **end function**
  $\tilde{\mathbf{a}}_{ij} \leftarrow$ SphericalHarmonicEmbedding($\mathbf{x}_{ij}$)      ▷ Spherical harmonic embedding of $\mathbf{x}_{ij}$ (Eq. (4))
  $\tilde{\mathbf{h}}_{ij} \leftarrow \tilde{\mathbf{f}}_i \oplus \tilde{\mathbf{f}}_j \oplus \|\mathbf{x}_{ij}\|^2$      ▷ Concatenate input for messages between $\tilde{\mathbf{f}}_i, \tilde{\mathbf{f}}_j$
  $\tilde{\mathbf{m}}_{ij} \leftarrow$ O3_TENSOR_PRODUCT_SWISH_GATE($\tilde{\mathbf{h}}_{ij}, \tilde{\mathbf{a}}_{ij}$)      ▷ First non-linear message layer
  $\tilde{\mathbf{m}}_{ij} \oplus \mathbf{g}_{ij} \leftarrow$ O3_TENSOR_PRODUCT($\tilde{\mathbf{m}}_{ij}, \tilde{\mathbf{a}}_{ij}$)      ▷ Second linear message layer
  $\tilde{\mathbf{m}}_i \oplus \mathbf{g}_i \leftarrow \sum_j \tilde{\mathbf{m}}_{ij} \oplus \sum_j \mathbf{g}_{ij}$      ▷ Aggregate messages $\tilde{\mathbf{m}}_{ij}$ and scalar irreps $\mathbf{g}_{ij}$
  $\tilde{\mathbf{f}}_i' \leftarrow \tilde{\mathbf{f}}_i +$ Gate($\tilde{\mathbf{m}}_i$, Swish($\mathbf{g}_i$))      ▷ Transform nodes via gated non-linearities

---

**Algorithm 3** Code sketch of an SEGNN message passing layer. The SEGNN layer updates steerable node features $\tilde{\mathbf{f}}_i' \leftarrow \tilde{\mathbf{f}}_i$. The additional scalar irreps $\mathbf{g}_i$ are used to "gate" the non-zero order irreps as introduced in Weiler et al. (2018). A fully documented code for this layer will be applicable in our repo.

---

**Require:** $\tilde{\mathbf{f}}_i, \mathbf{x}_{ij}, \mathbf{v}_i^1, \mathbf{v}_i^2$ ▷ Steerable nodes $\tilde{\mathbf{f}}_i$, relative position vector $\mathbf{x}_{ij}$ between node $\tilde{\mathbf{f}}_i$ and node $\tilde{\mathbf{f}}_j$, geometric or physical quantities $\mathbf{v}_i^1, \mathbf{v}_i^2$ such as velocity, acceleration, spin, or force.

  **function** O3_TENSOR_PRODUCT(input1, input2)
      output $\leftarrow$ CGTensorProduct(input1, input2)      ▷ Apply CG tensor product following Eq. (6)
      output $\leftarrow$ output + bias             ▷ Add bias to zero order irreps
      return output
  **end function**
  **function** O3_TENSOR_PRODUCT_SWISH_GATE(input1, input2)
      output $\oplus \mathbf{g}_i \leftarrow$ O3_TENSOR_PRODUCT(input1, input2)      ▷ Output plus scalar irreps $\mathbf{g}_i$
      output$_{\text{gated}} \leftarrow$ Gate(output, Swish($\mathbf{g}_i$))      ▷ Transform output via gated non-linearities
      return output
  **end function**
  $\tilde{\mathbf{a}}_{ij} \leftarrow$ SphericalHarmonicEmbedding($\mathbf{x}_{ij}$)      ▷ Spherical harmonic embedding of $\mathbf{x}_{ij}$ (Eq. (4))
  $\tilde{\mathbf{v}}_i^1 \leftarrow$ SphericalHarmonicEmbedding($\mathbf{v}_i^1$)      ▷ Spherical harmonic embedding of $\mathbf{v}_i^1$ (Eq. (4))
  $\tilde{\mathbf{v}}_i^2 \leftarrow$ SphericalHarmonicEmbedding($\mathbf{v}_i^2$)      ▷ Spherical harmonic embedding of $\mathbf{v}_i^2$ (Eq. (4))
  $\tilde{\mathbf{a}}_i \leftarrow \sum_j \tilde{\mathbf{a}}_{ij} + \tilde{\mathbf{v}}_i^1 + \tilde{\mathbf{v}}_i^2$      ▷ Node attributes
  $\tilde{\mathbf{h}}_{ij} \leftarrow \tilde{\mathbf{f}}_i \oplus \tilde{\mathbf{f}}_j \oplus \|\mathbf{x}_{ij}\|^2$      ▷ Concatenate input for messages between $\tilde{\mathbf{f}}_i, \tilde{\mathbf{f}}_j$
  $\tilde{\mathbf{m}}_{ij} \leftarrow$ O3_TENSOR_PRODUCT_SWISH_GATE($\tilde{\mathbf{h}}_{ij}, \tilde{\mathbf{a}}_{ij}$)      ▷ First non-linear message layer
  $\tilde{\mathbf{m}}_{ij} \leftarrow$ O3_TENSOR_PRODUCT_SWISH_GATE($\tilde{\mathbf{m}}_{ij}, \tilde{\mathbf{a}}_{ij}$)      ▷ Second non-linear message layer
  $\tilde{\mathbf{m}}_i \leftarrow \sum_j \tilde{\mathbf{m}}_{ij}$      ▷ Aggregate messages $\tilde{\mathbf{m}}_{ij}$
  $\tilde{\mathbf{f}}_i' \leftarrow$ O3_TENSOR_PRODUCT_SWISH_GATE($\tilde{\mathbf{f}}_i \oplus \tilde{\mathbf{m}}_i, \tilde{\mathbf{a}}_i$)      ▷ First non-linear node update layer
  $\tilde{\mathbf{f}}_i' \leftarrow \tilde{\mathbf{f}}_i +$ O3_TENSOR_PRODUCT($\tilde{\mathbf{f}}_i', \tilde{\mathbf{a}}_i$)      ▷ Second linear node update layer

---

## C.2 EXPERIMENTAL DETAILS

**Compared methods.** We compared against methods discussed in Sec. 3 of the main paper, namely NMP (Gilmer et al., 2017), SchNet (Kristof et al., 2017), Cormorant (Anderson et al., 2019), L1Net (Miller et al., 2020), LieConv (Finzi et al., 2020), TFN (Thomas et al., 2018), SE(3)-transformer (Fuchs et al., 2020), and EGNN (Satorras et al., 2021). We additionally compare to CGCNN (Xie & Grossman, 2018), PaiNN (Schütt et al., 2021), Dimenet++ (Klicpera et al., 2020) and SphereNet (Liu et al., 2021).

**Implementation details for N-body dataset.** SEGNN architectures, point conv methods (SE$_{\text{linear}}$), and steearable non-linear point conv methods (SE$_{\text{non-linear}}$) consist of three parts (sequentially applied):

1. Embedding network: Input $\to$ {CG$_{\tilde{\mathbf{a}}_i}$ $\to$ SwishGate $\to$ CG$_{\tilde{\mathbf{a}}_i}$}, wherein CG$_{\tilde{\mathbf{a}}_i}$ denotes a steerable linear layer conditioned on node attributes $\tilde{\mathbf{a}}_i$ and which are applied per node.

2. Four message passing layers as described in Sec. 2.1 for SEGNN architectures and 7 convolution layers for SE$_{\text{linear}}$ and SE$_{\text{non-linear}}$ ablations.

3. A prediction network: {CG$_{\tilde{\mathbf{a}}_i}^0$ $\to$ Swish $\to$ Linear $\to$ MeanPool $\to$ Linear $\to$ Swish $\to$ Linear}, in which CG$_{\tilde{\mathbf{a}}_i}^0$ denotes a steerable linear layer conditioned on node attributes $\tilde{\mathbf{a}}_i$ and which maps to a vector of 64 scalar (type-0) features. The remaining layers are regular linear layers, as in Satorras et al. (2021). CG$_{\tilde{\mathbf{a}}_i}^0$ and Linear are applied per node.

   All steerable architectures are designed such that the parameter budget at $l_f = 1$ and $l_a = 1$ matches that of the tested EGNN implementation. We optimise models using the Adam optimiser (Kingma & Ba, 2014) with learning rate 1e-4, weight decay 1e-8 and batch size 100 for 10000 epochs and minimise the mean absolute error. .

The SwishGate refers to a gated non-linearity (Weiler et al., 2018) when $l > 0$, and a swish activation (Ramachandran et al., 2017) otherwise. A message network $\phi_m$ consists of {CG$_{\tilde{\mathbf{a}}_{ij}}$ $\to$ SwishGate $\to$ CG$_{\tilde{\mathbf{a}}_{ij}}$ $\to$ SwishGate}, while the update network $\phi_f$ consists of {CG$_{\tilde{\mathbf{a}}_i}$ $\to$ SwishGate $\to$ CG$_{\tilde{\mathbf{a}}_i}$ $\to$ InstanceNorm}, with an instance normalisation (Ulyanov et al., 2016) implementation that is compatible with steerable vectors, based on the BatchNorm implementation of the `e3nn` library (Geiger et al., 2021a). In the default setting, however, the instance normalisation is turned off. We use skip-connections in the message passing layers, adapting Eq. (8) to yield $\tilde{\mathbf{f}}_i' = \tilde{\mathbf{f}}_i + \phi_f(\tilde{\mathbf{f}}_i, \sum_j \tilde{\mathbf{m}}_{ij}, \tilde{\mathbf{a}}_i)$.

The convolution layers which are used instead of the message passing layers for SE$_{\text{linear}}$ and SE$_{\text{non-linear}}$ ablations consist of:

- SE$_{\text{linear}}$: {CG$_{\tilde{\mathbf{a}}_{ij}}$ $\to$ $\sum_{j \in \mathcal{N}(i)}$ $\to$ SwishGate $\to$ InstanceNorm }
- SE$_{\text{non-linear}}$: {CG$_{\tilde{\mathbf{a}}_{ij}}$ $\to$ SwishGate $\to$ CG$_{\tilde{\mathbf{a}}_{ij}}$ $\to$ $\sum_{j \in \mathcal{N}(i)}$ $\to$ SwishGate $\to$ InstanceNorm}

Table C.1: A full ablation study on the performance and runtime of different (maximum) orders of steerable features ($l_f$) and attributes ($l_a$). The ablation study includes steerable linear convolutions, steerable non-linear convolutions, as well as SEGNN approaches. $SE_{non-linear}$++ uses the structural information (velocity) in the two embedding and read-out layers. Performance is measure using the Mean Squared Error (MSE).

| Method | MSE | Time [s] |
|---|---|---|
| $SE_{linear}$ ($l_f = 0, l_a = 0$) | .0344 | .003 |
| $SE_{linear}$ ($l_f = 0, l_a = 1$) | .0352 | .003 |
| $SE_{linear}$ ($l_f = 1, l_a = 1$) | .0130 | .031 |
| $SE_{linear}$ ($l_f = 1, l_a = 2$) | .0121 | .035 |
| $SE_{linear}$ ($l_f = 2, l_a = 2$) | .0116 | .065 |
| $SE_{linear}$ ($l_f = 2, l_a = 3$) | .0121 | .075 |
| $SE_{non-linear}$ ($l_f = 0, l_a = 0$) | .0382 | .004 |
| $SE_{non-linear}$ ($l_f = 0, l_a = 1$) | .0388 | .004 |
| $SE_{non-linear}$ ($l_f = 1, l_a = 1$) | .0061 | .030 |
| $SE_{non-linear}$ ($l_f = 1, l_a = 2$) | .0060 | .034 |
| $SE_{non-linear}$ ($l_f = 2, l_a = 2$) | .0061 | .065 |
| $SE_{non-linear}$ ($l_f = 2, l_a = 3$) | .0060 | .074 |
| $SE_{non-linear}$++ ($l_f = 0, l_a = 0$) | .0392 | .004 |
| $SE_{non-linear}$++ ($l_f = 0, l_a = 1$) | .0390 | .004 |
| $SE_{non-linear}$++ ($l_f = 1, l_a = 1$) | .0057 | .031 |
| $SE_{non-linear}$++ ($l_f = 1, l_a = 2$) | .0057 | .036 |
| $SE_{non-linear}$++ ($l_f = 2, l_a = 2$) | .0058 | .067 |
| $SE_{non-linear}$++ ($l_f = 2, l_a = 3$) | .0062 | .078 |
| $SEGNN_G$ ($l_f = 0, l_a = 0$) | .0099 | .004 |
| $SEGNN_G$ ($l_f = 0, l_a = 1$) | .0098 | .005 |
| $SEGNN_G$ ($l_f = 1, l_a = 1$) | .0056 | .025 |
| $SEGNN_G$ ($l_f = 1, l_a = 2$) | .0058 | .031 |
| $SEGNN_G$ ($l_f = 2, l_a = 2$) | .0060 | .061 |
| $SEGNN_G$ ($l_f = 2, l_a = 3$) | .0062 | .068 |
| $SEGNN_{G+P}$ ($l_f = 0, l_a = 0$) | .0102 | .004 |
| $SEGNN_{G+P}$ ($l_f = 0, l_a = 1$) | .0096 | .005 |
| $SEGNN_{G+P}$ ($l_f = 1, l_a = 1$) | .0043 | .026 |
| $SEGNN_{G+P}$ ($l_f = 1, l_a = 2$) | .0044 | .032 |
| $SEGNN_{G+P}$ ($l_f = 2, l_a = 2$) | .0041 | .063 |
| $SEGNN_{G+P}$ ($l_f = 2, l_a = 3$) | .0041 | .071 |

Table C.2: SEGNN and EGNN performance comparison on the N-body system experiment. Performance is compared for different number of available training samples, and measured using the Mean Squared Error (MSE). SEGNNs are significantly more data efficient.

| Method | Training samples | MSE |
|---|---|---|
| $SEGNN_{G+P}$ ($l_f = 1, l_a = 1$) | 1000 | .0068± .00023 |
| $SEGNN_{G+P}$ ($l_f = 1, l_a = 1$) | 3000 | .0043± .00015 |
| $SEGNN_{G+P}$ ($l_f = 1, l_a = 1$) | 10000 | .0037± .00014 |
| EGNN | 1000 | .0094± .00035 |
| EGNN | 3000 | .0070± .00022 |
| EGNN | 10000 | .0048± .00018 |

**Gravitational N-body dataset.** The gravitational 100-body particle system experiment is a self-designed toy experiment, which is conceptually similar to the charged N-body experiment described in the main paper. However, we extend the number of particles from 5 to 100, use gravitational interaction, no boundary conditions. Particle positions are initialised from a unit Gaussian, particle velocities are initialised with norm equal to one and random direction and particle mass is set to one. Trajectories are generated with by integrating gravitational acceleration in natural units using the leapfrog integrator with $\delta t = 0.001$ for 5000 steps. Force smoothing of 0.1 is applied (Dehnen & Read, 2011).

We predict position or force at step $t = 4$ given the state at $t = 3$. Such problems are often encountered in physical and astrophysical simulations (Alvaro et al., 2020; Tobias et al., 2020; Mayr et al., 2021), where usually the force (or acceleration) is used to Euler-update the positions. Thus, reliable force prediction is of great interest. An exemplary trajectory is shown in Fig. C.1. For the following experiments, we use 10.000 simulation trajectories in the training set, and 2.000 trajectories for the validation and test set, respectively. The implemented architectures are 4 layer graph neural networks with roughly equal parameter budget, and we train each network for 250 epochs. The SEGNN layers are exactly the same as described above for the charged N-body dataset, the EGNN and MPNN layers are the same as described in Satorras et al. (2021).

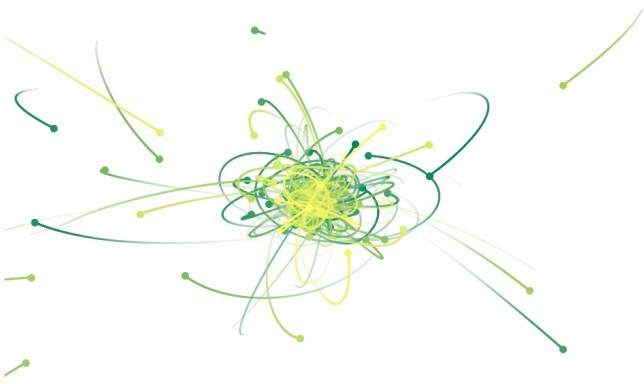

Figure C.1: Trajectory between $t = 4$ and $t = 5$ of 100 particles under gravitational interaction. Marker shows final position at end of simulation, with opacity decreasing over time.

Table C.3 shows results for different numbers of considered interactions (neighbours) for SEGNNs, as well as for message passing networks (MPNNs) and EGNNs. For SEGNN implementation, we input the relative position to the center of the system and the velocity as vectors of type $l = 1$ with odd parity. We further input the norm of the velocity as scalar, which altogether results in an input vector $\tilde{\mathbf{h}} \in V_0 \oplus V_1 \oplus V_1$. The output is embedded as difference vector to the initial position (odd parity) or the directly predicted force after 1.000 timesteps, i.e. $\tilde{\mathbf{o}} \in V_1$. In doing so, we keep E(3) equivariance for vector valued inputs and outputs. The edge attributes are obtained via the spherical harmonic embedding of $\mathbf{x}_j - \mathbf{x}_i$ as described in Eq. 4. The node attributes comprise the sum of the edge attributes plus the spherical harmonic embedding of the velocity. The MPNNs take the position and the velocity as input. Due to the vector-valued inputs and outputs, equivariance is not preserved. For EGNNs, equivariance is only preserved for the prediction of future positions, for the force prediction it is not.

MPNNs and SEGNN($l_f = 0, l_a = 0$) are conceptually very similar. Interestingly, MPNNs, which use PyTorch's Linear operation, are approximately 5 times faster than our SEGNN implementation, which uses Einsum operations. In general, the position and force prediction for the 100-body problem is intrinsically very difficult due to the wide range of different dynamics, where the main error contributions arise from a few outlier trajectories. However, aside from unavoidable errors,

SEGNNs generalise much better than default MPNNs or EGNNs do. The results strongly suggest the applicability of SEGNNs to large physical (simulation) datasets.

Table C.3: Mean Squared Error (MSE) for positional (pos) and force prediction in the gravitational N-body system experiment, and forward time in seconds for a batch size of 20 samples running on an NVIDIA GeForce RTX 3090 GPU.

| Method | 5 neighbours | | | 20 neighbours | | | 50 neighbours | | |
|---|---|---|---|---|---|---|---|---|---|
| | pos | force | Time[s] | pos | force | Time[s] | pos | force | Time[s] |
| MPNN | .297 | .299 | .0012 | .277 | .273 | .0014 | .262 | .268 | .0029 |
| EGNN | .301 | unstable | .0024 | .256 | unstable | .0025 | .239 | unstable | .0047 |
| SEGNN($l_f = 0, l_a = 0$) | .292 | .296 | .0085 | .266 | .276 | .0088 | .251 | .265 | .0100 |
| SEGNN($l_f = 1, l_a = 1$) | .265 | .273 | .0208 | .237 | .244 | .0212 | .212 | .223 | .0416 |

**Implementation details for QM9.** Our QM9 hyperparameters strongly resemble those found in (Satorras et al., 2021), whose architecture we took as a starting point for our own. The network consists of three parts (sequentially applied):

1. Embedding network: Input $\rightarrow \{\mathrm{CG}_{\tilde{\mathbf{a}}_i} \rightarrow \mathrm{SwishGate} \rightarrow \mathrm{CG}_{\tilde{\mathbf{a}}_i}\}$, wherein $\mathrm{CG}_{\tilde{\mathbf{a}}_i}$ denotes a steerable linear layer conditioned on node attributes $\tilde{\mathbf{a}}_i$ and which are applied per node.

2. Seven message passing layers as described in Sec. 2.1.

3. A prediction network: $\{\mathrm{CG}^0_{\tilde{\mathbf{a}}_i} \rightarrow \mathrm{Swish} \rightarrow \mathrm{Linear} \rightarrow \mathrm{MeanPool} \rightarrow \mathrm{Linear} \rightarrow \mathrm{Swish} \rightarrow \mathrm{Linear}\}$, in which $\mathrm{CG}^0_{\tilde{\mathbf{a}}_i}$ denotes a steerable linear layer conditioned on node attributes $\tilde{\mathbf{a}}_i$ and which maps to a vector of 128 scalar (type-0) features. The remaining layers are regular linear layers, as in Satorras et al. (2021). $\mathrm{CG}^0_{\tilde{\mathbf{a}}_i}$ and Linear are applied per node.

Message network, update network and skip-connections are used as described for the N-body task. In contrast to the other tasks, the hidden irreps correspond to $N$ copies for every order, selected to correspond to a 128-dimensional linear layer in terms of number of parameters.

We optimise models using the Adam optimiser (Kingma & Ba, 2014) with learning rate 5e-4, weight decay 1e-8 and batch size 128 for 1000 epochs and minimise the mean absolute error. The learning rate is reduced by a factor of ten at 80% and 90% of training. Targets are normalised by subtracting the mean and dividing by the mean absolute deviation. We found that pre-processing the molecular graphs for a specific cutoff radius on QM9 was beneficial.

Figure C.2 shows the number of messages as a function of cutoff radius for the train partition of QM9. It shows that a network with cutoff radius of 2Å sends $\sim 6\times$ fewer messages on average in every layer compared to a cutoff radius of 5Å, with far smaller variance.

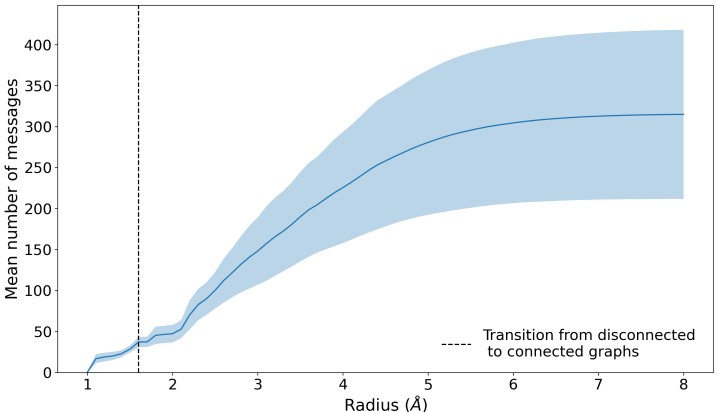

Figure C.2: Mean number of messages as a function of cutoff radius for the QM9 train partition. Standard deviation is indicated by the shaded region. The dashed black line indicates the transition from disconnected to connected graphs and may be thought of as a minimum cutoff radius.

**Implementation details for the Open Catalyst Project OC20 dataset.** Our OCP hyperparameters resemble those from the N-body task, but with an overall larger network architecture. The network consists of three parts (sequentially applied):

1. Embedding network: Input $\rightarrow \{\mathrm{CG}_{\tilde{\mathbf{a}}_i} \rightarrow \mathrm{SwishGate} \rightarrow \mathrm{CG}_{\tilde{\mathbf{a}}_i}\}$, wherein $\mathrm{CG}_{\tilde{\mathbf{a}}_i}$ denotes a steerable linear layer conditioned on node attributes $\tilde{\mathbf{a}}_i$ and which are applied per node.

2. Seven message passing layers as described in Sec. 2.1.

3. A prediction network: $\{\mathrm{CG}^0_{\tilde{\mathbf{a}}_i} \rightarrow \mathrm{Swish} \rightarrow \mathrm{Linear} \rightarrow \mathrm{MeanPool} \rightarrow \mathrm{Linear} \rightarrow \mathrm{Swish} \rightarrow \mathrm{Linear}\}$, in which $\mathrm{CG}^0_{\tilde{\mathbf{a}}_i}$ denotes a steerable linear layer conditioned on node attributes $\tilde{\mathbf{a}}_i$ and which maps to a vector of 256 scalar (type-0) features. The remaining layers are regular linear layers, as in (Satorras et al., 2021). $\mathrm{CG}^0_{\tilde{\mathbf{a}}_i}$ and Linear are applied per node.

Message network, update network and skip-connections are used as described for the QM9 tasks. The input is a 256-dimensional one-hot representation of atom types, all hidden vectors are 256-dimensional or their steerable "weight balanced" equivalent (cf Sec. 4), where we divide into equally large type-$l$ sub-vector spaces.

We optimise models using the AdamW optimiser (Loshchilov & Hutter, 2017) with learning rate 1e-4, weight decay 1e-8 and batch size 8 for 20 epochs and minimise the mean absolute error. The learning rate is reduced by a factor of ten at 50% and 75% of training.

**Ablation study for the Open Catalyst Project OC20 dataset, speed comparison.** Table C.1 shows SEGNN ablation of the performance when using different (maximum) orders of steerable features $l_f$ and attributes $l_a$, ablation for different cutoff radii of 5Å and 3Å, as well as runtime comparisons. All results are obtained on the validation set. The case $l_a = 0$ and $l_f = 0$ corresponds to our EGNN implementation. The SE$_{\text{linear}}$ corresponds to our implementation as outlined in Alg. 1 and can be seen as instance of e.g. TFNs Thomas et al. (2018). All methods have been optimised using roughly the same parameter budget as SEGNNs.

Table C.4: Comparison on the validation sets of the OC20 IS2RE tasks. **Model selection** is done on those datasets which are conceptually the nearest. We compare the performance for the usage of different (maximum) orders of steerable features ($l_f$) and attributes ($l_a$), as well as the performance for cutoff radii of 5Å and 3Å on the OC20 dataset. The case $l_a = 0$ and $l_f = 0$ corresponds to our EGNN implementation. The SE$_{\text{linear}}$ corresponds to our implementation as outlined in Alg. 1 and can be seen as instance of e.g. TFNs Thomas et al. (2018). Runtime numbers are reported for one forward pass and are measured for a batch of size 8 on a Nvidia Tesla V100 GPU. Higher orders of $l_f$ and $l_a$ in general do not improve results. The Dimenet++ runtime is given as reference.

| Model | Energy MAE [eV] ↓ | | | | EwT ↑ | | | | Time [s] |
| | ID | OOD Ads | OOD Cat | OOD Both | ID | OOD Ads | OOD Cat | OOD Both | - |
|---|---|---|---|---|---|---|---|---|---|
| EGNN($l_f = 0, l_a = 0, 5$Å) | 0.5497 | 0.6851 | 0.5519 | 0.6102 | 4.99% | 2.50% | 4.71% | 2.88% | 0.025 |
| EGNN($l_f = 0, l_a = 0, 3$Å) | 0.5931 | 0.7411 | 0.6182 | 0.6564 | 4.87% | 2.48% | 4.34% | 2.63% | 0.015 |
| SE$_{\text{linear}}$($l_f = 1, l_a = 1, 5$Å) | 0.5841 | 0.7655 | 0.6358 | 0.7001 | 4.32% | 2.51% | 4.55% | 2.66% | 0.028 |
| SE$_{\text{linear}}$($l_f = 1, l_a = 1, 3$Å) | 0.6020 | 0.7991 | 0.6631 | 0.7104 | 4.27% | 2.29% | 4.12% | 2.56% | 0.022 |
| SEGNN($l_f = 1, l_a = 1, 5$Å) | **0.5310** | 0.6432 | 0.5341 | **0.5777** | 5.32% | 2.80% | **4.89%** | **3.09%** | 0.080 |
| SEGNN($l_f = 1, l_a = 1, 3$Å) | 0.5523 | 0.6761 | 0.5609 | 0.6127 | 5.03% | 2.59% | 4.41% | 2.70% | 0.040 |
| SEGNN($l_f = 1, l_a = 2, 5$Å) | 0.5337 | **0.6419** | 0.5389 | 0.5764 | **5.41%** | 2.78% | **4.89%** | **3.09%** | 0.112 |
| SEGNN($l_f = 1, l_a = 2, 3$Å) | 0.5497 | 0.6662 | 0.5587 | 0.6060 | 4.98% | 2.51% | 4.40% | 2.60% | 0.051 |
| SEGNN($l_f = 2, l_a = 2, 3$Å) | 0.5369 | 0.6473 | **0.5335** | 0.5822 | 5.22% | **2.82%** | 4.87% | 2.98% | 0.234 |
| Dimenet++ | - | - | - | - | - | - | - | - | 0.062 |

# D LICENSES

This work would not have been possible without the existence of free software. As we have built extensively on existing assets, we mention them and their licenses here.

Our codebase uses Python under the PSF License Agreement, using NumPy under the BSD 3-Clause "New" or "Revised" License. Our models were implemented in PyTorch (Paszke et al., 2019) (BSD license) using CUDA (proprietary license). We used PyTorch extensions such as PyTorch Geometric (Fey & Lenssen, 2019) (MIT license) and e3nn (Geiger et al., 2021b) (MIT license). For tracking runs we used Weights & Biases (Biewald, 2020) (MIT license).

The QM9 (Ramakrishnan et al., 2014; Ruddigkeit et al., 2012) dataset was used under CC0 1.0 Universal (CC0 1.0) Public Domain Dedication and OC20 (Chanussot et al., 2020) was used under a Creative Commons Attribution 4.0 License. The QM9 dataloaders were adapted from (Anderson et al., 2019) under an Educational and Not-for-Profit Research License.

