# OpenReview forum: "Geometric and Physical Quantities improve E(3) Equivariant Message Passing"
_ICLR.cc/2022/Conference — ICLR 2022 Spotlight_

### Official Review · Reviewer_Mqkb · 2021-11-01

**Correctness:** 4
**Technical Novelty And Significance:** 4
**Empirical Novelty And Significance:** 4
**Recommendation:** 10
**Confidence:** 5

**Main Review:**

In my opinion, this is an outstanding paper that synthesizes an exciting area of recent work into a relatively authoritative general framework. I found the exposition to be extremely clear, especially given the technical nature of the subject matter. I also really liked the related works section that did a great job of positioning this paper in the context of the literature. Finally, the experimental results and ablation studies are thorough and seem to show a significant advantage for equivariant networks.

To summarize what I thought were the strengths of the paper:

1. The exposition was very clear throughout, but especially in the description of the model, the introduction of steerable vector spaces, the affine transformations, and related work. The supplementary material was also very thorough and contained a lot of really interesting and useful information. I would almost encourage the authors to publish a longer form version of this paper with everything in the main text.

2. I really liked the identification of the direct sum of steerable vector spaces (including multiplicity) with a type system that makes it straight-forward to add new physical quantities.

3. The formulation seems very general; while the authors apply it to MPNNs as defined in Gilmer et al. it seems like you could plug in the equivariant MLPs into any graph-network architecture.

4. The included code looks relatively clean; do the authors plan to open source it?

5. The empirical results seem very strong and three pretty different datasets. The Open Catalyst results are particularly nice to see, since the dataset seems much harder / larger than QM9 or the N-body system.

6. The ablations seem pretty thorough and it’s nice to see that equivariance seems to improve results of the l=0 (invariant) baseline.

While I did really like this paper, I do think there are a few places where it could be a bit more clear.

1. Although most of the design decisions seemed quite reasonable, I thought the paper could do a better job describing how the conditioning vectors, a, were chosen. Are they always the node or edge features? For the N-body system, were the conditioning vectors the velocity vector or the displacement vector to the center of the system? Did you try any other choices?

2. I would include a description of the gated nonlinearities in the SI of the paper (if not also in the main text). I would also be a little bit more straightforward about the fact that you use gated non-linearities. In particular, in the text you currently write [1] but in the text you only ever use gated non-linearities. I would lead with the choice you made and then mention that one could use any equivariant MLP.

3. One of the things I liked about this paper was how straightforward the proposed equivariant MLPs are. The submitted code uses e3nn, which (while very nice) is a pretty complex codebase. Can the authors discuss this choice vs writing things from scratch using Clebsch-Gordan coefficients as described in the paper?

4. Do the authors ever use the fact that the weights can be made dependent on a parameter? I couldn't find any place in the paper where this was used (perhaps I missed it), but if it's not used then I'm not sure it adds much to the discussion.

[1] _The common recipe for deep neural networks is to alternate linear layers with element-wise non-linear activation functions. In the steerable setting, careful consideration is required to ensure that the activation functions are equivariant; currently available classes of activations include Fourier-based (Cohen et al., 2018), norm-altering (Thomas et al., 2018), or gated non-linearities (Weiler et al., 2018). All these can be used in alternation with (6). The resulting steerable MLPs themselves in turn provide a new class of steerable activation functions that is able to directly leverage local geometric cues. Namely, through steerable node attributes a, either derived from the physical setup (forces, velocities) or from predictions (similar to gating), the MLPs can be applied node-wise and generally used in steerable feature fields as non-linear activations._


**Summary Of The Paper:**

Motivated by a wide array of applications across the physical sciences, there has recently been a flurry of interest in building neural networks that are equivariant to the symmetries of the Euclidean group in three-dimensions. In my opinion, this paper presents a particularly elegant and general framework for building equivariant networks that generalizes a number of recent results. There are two essential ingredients: 1) steerable linear transformations and 2) steerable nonlinearities. Both pieces are defined over direct sums of steerable vector spaces and, with these two ingredients, it is easy to promote any message-passing neural network to an equivariant one. By describing the problem in terms of direct sums of steerable vector spaces, there is sort of a nice “type checking” nature when wanting to include new geometric quantities.

**Summary Of The Review:**

A very strong paper that provides an elegant framework for building E(3) equivariant graph networks. The proposal is very general and well-motivated; it is also well-situated within the literature. The exposition is generally high quality, and the experiments are rigorous and compelling. The supplementary material gives thorough background and the code seems pretty clean. A few comments / questions about some details of the paper remain outstanding.

---

> ### Author Response · Authors · 2021-11-18
> **Response to Reviewer Mqkb**
>
> Thank you very much for this extremely encouraging review and your comments, which further helped to improve our paper.
>
> While we answer the individual points inline we want to stress one point in advance.
> `I really liked the identification of the direct sum of steerable vector spaces (including multiplicity) with a type system that makes it straight-forward to add new physical quantities.`
> We thank the reviewer to have explicitly pointed that out. As written in the general response and as response to other reviewers as well, we want to highlight that we explicitly strive to present our method in connection to as many methods as possible without obfuscation or additional complexity. It turned out that non-linear steerable updates conditioned on geometric and physical quantities is unique and has not been done before.
>
> * We will release our codebase with many exemplary use cases upon publication. We will also release the new datasets.
>
>
> * `Although most of the design decisions seemed quite reasonable, I thought the paper could do a better job describing how the conditioning vectors, a, were chosen. Are they always the node or edge features? For the N-body system, were the conditioning vectors the velocity vector or the displacement vector to the center of the system? Did you try any other choices?`
> Fully agreed. We have added pseudocode in Appendix C1 which should in general make the composition of the different layers more applicable, and furthermore give a more concrete description of how to choose the conditioning vectors.
> For the N-body system, the relative position vectors are used as steerable edge attributes. The sum of relative position vectors is used as steerable node attribute. Additionally, the velocity given at each node is added to the node attributes. This results in a significant performance improve, and physically makes sense since for position updates the velocity is the most important quantity and thus it’s equivariant injection at every node update surely helps. Other tested choices were to add relative velocities to the edge attributes, calculate electromagnetic forces and add forces (between node i and j) to the edge attributes, as well as summed relative forces to the node attributes (summation over neighbouring relative forces). In general, it is to say that there is a strong correlation between physical plausible choices and reflected results. For example adding relative and summed forces at some point slightly improved results. We have however chosen not to show these results since in our opinion it obfuscates the presentation.
>
>
> * `I would include a description of the gated nonlinearities in the SI of the paper (if not also in the main text). I would also be a little bit more straightforward about the fact that you use gated non-linearities. In particular, in the text you currently write [1] but in the text you only ever use gated non-linearities. I would lead with the choice you made and then mention that one could use any equivariant MLP.`
> Thank you very much for pointing this out. We have slightly changed the text accordingly. We have further taken care to correctly introduce the gated non-linearities in our added pseudocode. Lastly, we have added a short description of the working principle of gated non-linearities into the appendix C.1.
>
> * `One of the things I liked about this paper was how straightforward the proposed equivariant MLPs are. The submitted code uses e3nn, which (while very nice) is a pretty complex codebase. Can the authors discuss this choice vs writing things from scratch using Clebsch-Gordan coefficients as described in the paper?.`
> The e3nn implementation is based on einsum, applying these per path. We have tried an implementation that directly uses multiplications and summations in PyTorch, since einsum has been known to be slower than applying operations directly. This can be seen in table C3, where the MPNN, which uses PyTorch’s Linear operation, is approximately 5x faster than the SEGNN, which uses einsum. While our reimplementation yielded a speedup of approximately a factor 2 in small-scale tests, this implementation was slower when slotted into actual torch.nn modules. We are not fully certain as to why, but will be looking into this in follow-up work.
>
> * `Do the authors ever use the fact that the weights can be made dependent on a parameter? I couldn't find any place in the paper where this was used (perhaps I missed it), but if it's not used then I'm not sure it adds much to the discussion.`
> Yes, very good point. People using the e3nn library very often tend to do this. They use a single spherical harmonics (direction vector) as the filter and parameterise the weights using an mlp with basis functions and the distance (invariant) input. We have discussed this e.g. in Eq. (14). We have however not ablated it in our architectures. We see the CG tensor product more as direct mapping between vector spaces where the dependencies come as second input.

---

### Official Review · Reviewer_vbV1 · 2021-11-04

**Correctness:** 3
**Technical Novelty And Significance:** 3
**Empirical Novelty And Significance:** 3
**Recommendation:** 6
**Confidence:** 3

**Main Review:**

I have listed the contributions of this submission in the summary above. I will mainly focus on the questions below:

1.	This paper is not easy to follow. I understand that this is not owing the writing (the presentation is good anyway) but due to the abundant preliminaries to understand the geometry of steerable functions. The authors are suggested to separate the introductions of the preliminaries, related work, and the proposed methodology, instead of introducing them as a whole (section 2 and 3).

2.	The authors claim that the proposed SEGNN (Eq. (16)) is a general form of various previous steerable models (Eq. (14-15)). In my understanding, the main difference of SEGNN from previous steerable models (Thomas et al., 2018 and Fuchs et al., 2020) is that the hidden feature h in Eq. (16) is now a concatenation of the node steerable features and the squared relative distance. The idea of making the MLP (i.e. W) conditional on the attribute a_{ij} (e.g. be a spherical harmonic embedding of xj-xi) has been explored before. So, it seems the contribution is minor. Do I miss something? Perhaps, the authors should specify the form of each previous model under the same denotations and pipeline, from which we can easily check the difference between different models.

3.	 In terms of the nonlinearity, the authors claim that “Although powerful, such layers only (pseudo1) linearly transform the graphs and non-linearity is only obtained via point-wise activations.” However, in my opinion, Eq. (16) first computes the linear steerable convolution and then performs the nonlinear activation \sigma which is also point-wise. So, why we have the claim that the convolution is nonlinear? By the way, what is the form of \sigma? Does it directly follow the previous papers?

4.	I do not check the very details of the experimental part. Just justified from the performance, it seems on QM9, the performance between SEGNN and EGNN is close. Given that more complexity is involved in SEGNN for the computation of Spherical harmonic and Clebsch-Gordan products, the benefit of involving steerable features is not well verified experimentally. Besides, on OC20, there is no comparison with EGNN, why? Regarding the ablation studies, the difference between SEGNN-linear and -nonlinear is unclear. And it is suggested that the authors should compare their SEGNN with previous models under a more complete and fair setting. Since they are claimed as a specific case of SEGNN, the authors can keep other parts unchanged and check which component works actually.


**Summary Of The Paper:**

This paper introduces a kind of Steerable E(3) Equivariant Graph Neural Networks (SEGNNs), exhibiting these merits: 1) Compared to EGNN (Satorras et al., 2021) and other concurrent works, SEGNN does not restrict the message passing to be scalars but to be steerable features, and it also permits the injection of geometric and physical quantities into node updates. 2) Compared to previous steerable architectures (such as the works by Thomas et al., 2018 and Fuchs et al., 2020), SEGNN employs more general steerable convolutions, built upon the message passing network (Gilmer et al., 2017). The experiments are conducted on several benchmarks: N-body, QM9 and OC20.

**Summary Of The Review:**

I generally think the proposed general steerable GNN is meaningful. Yet, given certain unclarities related to the novelty and experimental significance, I initially suggest that it is just marginally abve the acceptance bar.

---

> ### Author Response · Authors · 2021-11-18
> **Response to Reviewer vbV1**
>
> Thank you very much for your detailed review. Let us first say, we are well aware that the topic of the paper has abundant preliminaries. We however see it as our duty to provide a self-consistent description including both mathematical background and related architectures. Especially for the latter a strong contribution of this paper is the unification of various (steerable) group convolution methods. It is exactly this unification which allows us to pinpoint important and successful concepts and further build on them. Regarding mathematical preliminaries, we try to provide a compact and  yet comprehensive description of those to help future research.
>
> **Novelty**: We are to the best of our knowledge the first method which both uses non-linear message and node updates and is able to incorporating geometric and physical quantities as steering features in both message and node updates. Especially, the steering of node updates has not been done before, and proves as a very powerful tool. The steering of the node updates can also be seen as new type of steerable activation function, the first activation function which includes geometric and physical priors. We have added pseudocode to the paper draft which should make the difference between linear messages, non-linear messages and SEGNNs (non-linear messages and non-linear node updates) clearer. Especially for the latter it becomes clear that only via the non-linear node updates one suddenly is offered the chance to include geometric and physical priors into the networks. We consider this as a very important feature, and have added another gravitational 100-body dataset to showcase this again.
> As written in the general response, we want to highlight that we explicitly strive to present our method in connection to as many methods as possible without obfuscation or additional complexity. Most notably, it is not possible in any of the related methods to incorporate geometric and physical quantities. Therefore modifications, although appearing small in a general context, make a big difference when processing physical (simulation) datasets.
>
> **Difference between linear and non-linear convolution and your comment regarding Eq. 16 of our draft**: This should hopefully be clarified by looking at the differences of Algorithm 1 and Algorithm 2. In the latter, a non-linearity is applied to each message before aggregation. The application of this non-linearity is a bit tricky however, since extra 0-order irreps need to be reserved for the gated non-linearities.
>
> **Performance comparison with EGNN**: We compare with EGNNs in every experiment. As noted correctly by the reviewer, this is our most important baseline.
> However, for the OC20 dataset the comparison is hidden in the appendix (Table C3) The reason is that for official results the number of submissions is limited, and thus we have to do comparisons on the validation sets. We have renamed SEGNN(lf=0,la=0) to EGNNs (our EGNN implementation) to make it clearer. We have further added our SE_linear baseline.
> We further fully agree with the statement that performance gains (between SEGNN and EGNN) are not overwhelming on the QM9 dataset. We want to remark that SEGNNs are designed for vector valued inputs and outputs, as well as available geometric and physical information. Only positional information is available for QM9. Datasets for which SEGNNs are particularly suited are physical and fluid simulation datasets. An example is the n-body dataset where we have highlighted the benefits of SEGNNs. For further comparison to EGNNs we have added another table (Table C2 in the appendix), where we compare SEGNN and EGNN performance for different number of training set sizes, with pretty convincing results. And finally, to stress this point even more, we have added another experiment to the paper draft with a 100-body gravitational dataset, highly relevant e.g. for astrophysics simulation. We want to stress that we did experiments especially on QM9 to have all these comparisons to existing methods even if it is not our number one choice dataset due to missing characteristics.

---

> > ### Comment · Reviewer_vbV1 · 2021-11-23
> > **Thanks for the response**
> >
> > I appreciate the author's efforts in the provided clarification. I have no question now and wait for the discussion with other reviewers.

---

### Official Review · Reviewer_aA3f · 2021-11-05

**Correctness:** 4
**Technical Novelty And Significance:** 2
**Empirical Novelty And Significance:** 2
**Recommendation:** 8
**Confidence:** 4

**Main Review:**

The starting point for this paper is the observation that regular MPNNs (of which CNNs can be seen to be a special case) get their power from the fact that node features (which are usually invariant scalars) are transformed and propagated in a non-linear manner. Work on equivariant GNNs  is more general, but they still linearly transform the graph and the non-linearity is only obtained via point-wise activations. The authors propose to generalize this formalism and propose E(3) equivariant graph networks, viewing the message passing phase itself as performing non-linear convolution. Further, covariant information is incorporated into the nodes and edges, and they also transform equivariantly. A major inspiration is a recent work by Satorras et al. Indeed, the paper can be seen building on it directly while adding in what they call "non-linear convolutions" and augmenting the nodes and edges to include covariant information.

To go into more detail about some of the above points: the usual message passing formalism is defined through equations (1) and (2). phi_m is the message function and phi_f is the aggregation/update function. phi_m and phi_f are required to be equivariant, specifically to the group E(3). Furthermore, physical and geometric information is incorporated in the node and edge attributes as steerable vectors which transform covariantly as the message passing propagates. For this, the authors define steerable MLPs on page 3 (same functional form as MLPs, but the weight matrices are conditioned on geometric information using the CG products, and thus support covariance). The exact form is encapsulated in equation (6). Section 2.1 is essentially the same scheme as in Satorras et al. but with the steerable MLP added in. Section 3 goes over the related work on viewing message passing as convolution. The usual convolution layers are discussed first (eq. 6), followed by their usual treatment in point clouds aka "point convolution." The important point is that the matrices W (the convolution kernel) are conditioned on relative positions. The exact form depends on the type of input data. The setup used in the paper corresponds to a steerable convolution given in equations (12) and (13) (the relative distances are embedded in the spherical harmonic space). Because of the CG products that W crucially relies on, the authors interpret their method as performing a non-linear convolution (the approach of Satorras can be seen as corresponding to the special case when the kernel has an isotropic form).

Quite simply, I see the paper as proposing an extension of the approach of Satorras. The message passing scheme is the same. The relative distances are embedded into the spherical harmonic space, the node and edge attributes are steerable, and the message passing ensures these are transformed covariantly. In this sense, I feel like the paper has limited novelty, or at least overstates its contributions (this happens frequently throughout the paper).

In the beginning one gets the impression that something new is going on with "non-linear convolutions", but it just amounts to taking CG products, an idea that has been used in other contexts quite successfully 2-3 years ago. It is also said that incorporating steerability into GNNs is a new contribution -- this is only true with respect to the work of Satorras which just passes invariant messages, which was indeed pleasantly surprising at the time in terms of performance and simplicity. For instance the paper "Covariant Compositional Networks for Learning Graphs.", Risi Kondor, Hy Truong Son, Horace Pan, Brandon Anderson, Shubhendu Trivedi, has considered this problem in detail. The case for steerability is mentioned in the abstract, and the intro. Besides, the "type 0" and "type k" message passing types are also discussed in detail later. The difference is that node and edge attributes are not required to be covariant with respect to some Euclidean group. However, the main idea there, although it just deals with the permutation group, is about steerability. One can replace the defining representation there with a representation for a group of interest, along with attaching steerable vectors on nodes and edges, and their approach will also work with the same import as reported in this paper. Papers such as "Vector Neurons: A General Framework for SO(3)-Equivariant Networks" Congyue Deng, Or Litany, Yueqi Duan, Adrien Poulenard, Andrea Tagliasacchi, Leonidas Guibas, are not cited and should be discussed properly, along with some related papers such as Jing et al. (2020). These are more related to this paper than the author let on.
Taking tensor products can also become prohibitive quite quickly, unless I missed it (and I am sorry if I did), I don't see a discussion about this in the main paper (nor about what L is considered and what effect it has on the runtime versus accuracy curve).

The experimental results reported in the paper are quite encouraging, and for me the hallmark of the paper.

Minor comment:
Page 6, below equation (15): "There are, however, two import differences" -> "There are, however, two important differences"

**Summary Of The Paper:**

The paper extends the recently proposed E(n) GNN (Satorras et al.) such that node and edge attributes can contain covariant information. This allows the easy incorporation of geometric and physical information in the MPNN framework in a steerable manner. Excellent results are reported across a set of benchmarks.

**Summary Of The Review:**

I agonized over the rating to give this paper. On one hand, it reports a clear and unambiguous benefit of incorporating covariant information in the node and edge attributes. It is also well-motivated and well written and is definitely valuable. However, the technical contribution of the paper at this point (in how the equivariant neural network literature is progressing) is relatively minor. The paper also seems to overclaim its contributions, one case being in the sense of how it relates to steerable graph networks (although this happens because they are positioning themselves in contrast to Satorras et al., Thomas et al., etc). While I am a big fan of simple methods that just work well, I feel conflicted about this paper. I am currently rating the paper a 6 (which is mostly due to their experimental results), I would be willing to adjust my rating if the authors can present arguments showing that my concerns are unfounded or not rational.

---

> ### Author Response · Authors · 2021-11-18
> **Response to Reviewer aA3f**
>
> Thank you very much for the insightful review.
>
> **Novelty of contribution**:
> Let me first say that we are sorry if we overstate the novelty of our paper in your opinion. We have already addressed the issue in the general response, but want to highlight again the most important points of our paper which we consider novel.
> First of all, there is an abundancy of SE(3) equivariant point conv and graph networks.
> We rigorously analyzed many of these methods and put them into a unified framing. We thereby identify steerable non-linear messages and steerable node updates as most important components. The latter makes it possible to inject geometric and physical quantities such as velocity or force into the node updates, and therefore can be seen as new type of steerable activation function. By doing this, we are to the best of our knowledge the first method which is able to incorporating geometric and physical quantities as steering features in both message and node updates. Especially, the steering of node updates has not been done before, and proves as a very powerful tool.
> The steering of the node updates can also be seen as new type of steerable activation function, the first activation function which includes geometric and physical priors.
>
> As written in the general response, we want to highlight that we explicitly strive to present our method in connection to as many methods as possible without obfuscation or additional complexity. Most notably, it is not possible in any of the related methods to incorporate geometric and physical quantities. Therefore modifications, although appearing small in a general context, make a big difference when processing physical (simulation) datasets.
>
> We have (i) added pseudocode for SEGNN architectures and two ablations.
> The non-linear mapping between steerable feature spaces and the accompanying injection of geometric and physical information such as velocity and force is highlighted for SEGNN architectures. We strongly believe that this is a very important general recipe for processing physical (simulation) data and have therefore have (ii) added a gravitational 100-body dataset to the paper draft.
>
> **Related work**:
> Concerning comp-nets (Kondor 2018), we did not mean to claim to be the first steerable GNN, merely the first general E(3) steerable GNN. This novelty lies in our investigation of non-linearly transforming local information before aggregation, which distinguishes message passing from convolution. As shown in the ablations on the N-body dataset, this non-linear convolution, as we termed it, improves performance significantly.
> As for Vector Neurons (Deng 2021), similar to Geometric Vector Perceptrons (Jing 2020), these methods omit certain types of information or specific interactions. For example, scalar features in GVP are updated using vector-valued features—but not vice versa. Similarly, Vector Neurons are incapable of incorporating scalar-valued information. Both methods—similarly to EGNN—omit the need for steerable features, but sacrifice some generality in doing this. Our ablations on the feature order l show that higher-order interactions are beneficial and suggest that higher-order interactions can be worth the incurred computational cost. We have already cited Jing 2020 and are happy to include Deng 2021 in our new paper draft.
>
> **Clebsch-Gordan tensor product**:
> Regarding the statement that the tensor product can can also become prohibitive quite quickly. We have ablated higher orders of the Clebsch-Gordan tensor product wherever possible. We have further added a Section into Appendix A.5 where we explain they reasons for the higher computational cost of the Clebsch-Gordan tensor product at higher orders.

---

> > ### Comment · Reviewer_aA3f · 2021-11-23
> > **Updates**
> >
> > I am not convinced about responses to some of the points I raised (especially with regard to placement of the work). However, I read all of the comments made rebutting each review, as well as the updated paper, and I think it would be fair to revise my score up one notch despite my misgivings.

---

### Official Review · Reviewer_k14d · 2021-11-08

**Correctness:** 3
**Technical Novelty And Significance:** 3
**Empirical Novelty And Significance:** 3
**Recommendation:** 6
**Confidence:** 3

**Main Review:**

### Strengths
- There are several existing equivariant graph neural networks including TFN, SE(3)-transformers, Schnet, and E(n)-Equivariant GNNs (EGNN).  I was thus a bit surprised to see there does not exist a full-power equivariant GNN in the popular message passing flavor (MPNNs).  By “full power” I mean able to handle a range of equivariant feature types and learn over a large space of interwiners between them.  Schnet and EGNNs are MPNNs which are technically equivariant but primarily handle scalar features (invariants) and to a limited extent vector features (standard rep) but without representing the full space of equivariant linear mappings.  The current work is thus novel and extends to a very reasonable combination of successful aspects since both MPNNs and equivariant networks with higher order features types have both proved successful in the past.
- In particular the current method uses anisotropic convolutions which is one of the primary advantages of uses non-invariant equivariant feature types.
- The method achieves competitive results with several baselines on an N-body problem and two molecular properties tasks.
- One of the more interesting aspects of the method is that convolution is taken not necessarily over the usual spatial base space.  The graphs here are embedded in R^3, however, the features may be considered as living over a higher dimensional state space consisting of position and velocity.  In physics, it is often more reasonable to compute over this higher dimensional space, and the author find that doing so improves their model.

### Weaknesses
- The paper suffers from some lack of precision and formalism.  The authors describe their method as “non-linear” convolution.  With respect to which definition of convolution?  Section 3, in particular is very difficult to read.  The first part appears to try to compare regular group convolutions and steerable group convolutions but without defining either or stating a precise theorem.
- Section 3 is also claimed as a contribution in the form of “unifying view.”  As it stands, I would not consider that to be the case.
- The authors claim, for example, that their method provides a new class of activation functions for equivariant NNs, but so far as I can tell, the non-linearity is comparable to TFN, this claim merely stems from considering a NN as a non-linearity. This does not seem like a useful extension of the terminology and obfuscates the contribution.

### Questions
- Why are TFN and EGNN not used as baselines for OC20?
- DimeNet, SphereNet, PaiNN are all shown with better numbers for QM9, but an asterisk notes they use a different set.  It is hard for me to know whether it is reasonable for me to use these numbers to compare.  Is the code or data not available to make a fair comparison?
- On page 7, it is noted that l_a = l_f = 1 usually worked best.  Isn’t this case already considered by SchNet and EGNN?

**Summary Of The Paper:**

This paper proposes SEGNNs, a message passing neural network which is equivariant to 3D rotations.  The message passing neural network is a type of graph neural network consisting of edge networks and node networks.  Equivariance is achieved by enforcing SO(3) equivariance on the node and edge networks using what the authors call Steerable MLPs.  Steerable MLPs use spherical harmonics and Clebsch-Gordon coefficients similar to Tensor Field Networks but without aggregation.  SEGNNs may be considered non-linear convolutional networks with equivariant anisotropic messages.  The base space for the convolution may be taken to be not just spatial, but a state space consisting of, e.g. position and velocity.  The method achieves competitive results with several baselines on an N-body problem and two molecular properties tasks.


**Summary Of The Review:**

Although parts of the paper were difficult to understand and some of the contributions are questionable, the core contribution, making equivariant GNNs in the message passing flavor is a useful step and together with considering higher-dimensional state spaces seems to result in improved performance.  Thus I tend towards accept, but I would prefer revisions.

---

> ### Author Response · Authors · 2021-11-18
> **Response to Reviewer k14d**
>
> We thank the reviewer for the comments and the time spent reviewing our paper.
>
> The first two bullets under weakness can be summarized as a lack of formalism for non-linear convolutions, and precision towards how it can be regarded as a unifying view. We address this concern by rewriting the intro of section 3:
>
> *Recent literature shows a trend towards building architectures that improve performance by means of maximally preserving equivariance through groups convolutions (either in regular or tensor-product form, see App. B). Convolutions, however, are ”just” linear operators and non-linearities are only introduced through point-wise activation functions. This is in contrast to architectures that are built without explicit use of group convolutions, but instead rely on the highly non-linear framework of message passing. In the following we show that many related works are connected through a notion of non-linear convolution, a term that we coin based on the following. Any linear operator which is equivariant is a group convolution (Kondor & Trivedi, 2018; Bekkers, 2019) and their discrete implementations can be written in message passing form. We then call any non-linear operator that is equivariant, and which can be written in simple message passing form, a non-linear convolution. This framing allows us to place related work in a unifying context and to identify two important aspects of successful architectures: (i) equivariant layers improve upon invariant ones and (ii) non-linear layers improve upon linear ones. Both come together in SEGNNs via steerable non-linear convolutions.*
>
> With this introductory statement we provide a slightly more formal introduction to the notion of non-linear convolutions, yet we refrain from the formal theorem-proof format as requested. We do so as we deem the current exposition more intuitive and accessible to a broader audience. We hope that this alleviates your concerns nevertheless. We further remark that regular and steerable group convolutions are discussed in App. B (we now included explicit reference in section 3) and that we now also discuss steerable group convolutions vs non-linear group convolutions on an algorithmic level via Pseudo-code, discussed in detail in the newly added App. C.1.
>
> Regarding the third bullet, this concern is related to our non-linear convolution perspective. Most of our insights come from connecting related works through the message passing framework and recognizing that (group) convolutional methods only have non-linearities in the update function. In TFN-type methods the non-linearity is typically a gated non-linearity (the most common and powerful activation for steerable methods). Our activation functions are steerable MLPs themselves, which themselves may use the classical activation functions (such as gating) in their layers, this is what we do. So yes, from one perspective this is “just” changing the point-wise gating with a point-wise application of a NN. However, up until now, one could not just apply NNs point-wise as one has to respect the steerability/equivariance constraints. Furthermore, in order to create sensible equivariant layers one needs two (not one) steerable vectors (for use in the CG-product). Through the introduction of steerable node attributes we can build steerable MLPs that can be applied point-wise.
>
> Note that, one could in principle build steerable MLPs without attributes, however, this would correspond to a kind of point-wise spherical CNNs that does not leverage any form of geometric or physical quantities otherwise. What we do in this paper is introducing steerable node attributes such that geometric and physical quantities can be exploited even in the node-wise updates!
>
>
> * `Why are TFN and EGNN not used as baselines for OC20?`
> Good catch. In Table C.4 SEGNN with la=0, lf=0 is an EGNN baseline. We renamed that properly. Please note that we can only do this comparisons on the validation set due to limited submission to the official challenge. We have further added a SE_linear model to Table C.4, which can be considered as TFN baseline.
>
> * `DimeNet, SphereNet, PaiNN are all shown with better numbers for QM9, but an asterisk notes they use a different set. It is hard for me to know whether it is reasonable for me to use these numbers to compare. Is the code or data not available to make a fair comparison?`
> Models with an asterisk use 110k samples for training whereas the other models (ours included) use 100k samples for training. We do not claim that the different training set sizes make a big difference, but consider it still important to note. We have chosen the same training set size as e.g. TFN, MPNN, EGNN since these are the models which we used to derive our SEGNN architecture.
>
> * `On page 7, it is noted that l_a = l_f = 1 usually worked best. Isn’t this case already considered by SchNet and EGNN?`
> SchNet and EGNN do not use higher order features, thus it would correspond to the case la=0 and lf=0

---

### Official Review · Reviewer_zF5H · 2021-11-08

**Correctness:** 4
**Technical Novelty And Significance:** 4
**Empirical Novelty And Significance:** 4
**Recommendation:** 6
**Confidence:** 3

**Main Review:**

This paper is well-motivated and well-written. I have few concerns below for the authors to address.

Minor concerns:
1. How to understand that using the spherical harmonic embedding of relative positions is including physical information? Is that a pure geometric information?
2. How to include relative force or relative momentum in the molecular tasks, e.g., the QM9 and OC20 experiments used in the paper.
3. I assume that EGNN should be an important baseline of this paper, but seems that EGNN has not been reported on OC20.
4. I'm very curious that why equivarience is important (or say, how to prove it) for the molecular tasks, e.g., QM9 and OC20, which induces a series of equivarient model architecture development. As far as I know, some methods which aren't equivarient perform well on similar tasks, e.g., Very Deep Graph Neural Networks Via Noise Regularisation, Godwin et al.
5. The performance of the proposed method is not very competitive. For example, UNiTE largely outperform baseline methods (SphereNet, PaiNN, etc) on QM9, but the performance gain of proposed SEGNN is marginal. Similarily, GemNet (Klicpera et al., NeurIPS 2021) and NoisyNode (Godwin et al.) perform well on OC20 IS2RE.
6. Considering that the recent GemNet is also a powerful equivarient GNN, would the authors make a more comprehensive discussion to explain the pros/cons between these two methods?
7. Equivarient model is a hot topic today in the community, and they could be developed by different techniques, e.g., integral (group equivarient CNN, spherical CNN,...), tensor product (TFN, SE(3)-Tr, ...), or equivarient coordinates (EGNN, AF2, ...). Would the authors explain more about how should we think about these equivarient models, how to justify whether an equivarient model is better than others, and what's the future directions of equivarient models?
8. I'm not very familiar with the mathmatics in this paper, so I could not guarantee all the theories are correct in the paper. I'm open to other reviewers opinions.


**Summary Of The Paper:**

This study presents a improved version of EGNN (E(n) Equivariant GNN, Victor et al. 2021), where the node features and edge attributes are replaced by steerable vectors. In addition, the MLPs in message passing should also be replaced by steerable MLPs. The advantage of steerable vectors on the node and edge is that attributes are not restricted to scalars but also can be covariant information (e.g., vectors, tensors). The experimental results on N-body, QM9 and OC20 demonstrate the effectiveness of proposed method.

**Summary Of The Review:**

Overall, I enjoy reading this paper, but I have minor concerns. I'm definitely willing to raise my score if my concerns are well addressed.

---

> ### Author Response · Authors · 2021-11-18
> **Response to Reviewer zF5H (2)**
>
> * `Equivarient model is a hot topic today in the community, and they could be developed by different techniques, e.g., integral (group equivarient CNN, spherical CNN,...), tensor product (TFN, SE(3)-Tr, ...), or equivarient coordinates (EGNN, AF2, ...). Would the authors explain more about how should we think about these equivarient models, how to justify whether an equivarient model is better than others, and what's the future directions of equivarient models?`
> We fully agree that this is a very important question, and therefore have dedicated a large section (Section 3 in the main paper and Appendix B) in our paper to discuss various equivariant models in a unified framework. It should be noted that every linear layer that is equivariant is a group convolution. This statement then connects the integral methods as well as the tensorproduct method above (discussed in Appendix B). Then there are also methods that are equivariant, but do not follow the standard convolution point-wise nonlinearity alternation, but are built on more intricate graph message passing schemes.
> In this paper we show that by means of equivariant message passing, one constructs powerful networks mainly through a notion of increased non-linear processing. We believe that the future of equivariant models is indeed increased non-linearity, whilst preserving equivariance as much as possible.
> For example, EGNN is only partially equivariant but is highly non-linear. Tensor product methods are mostly linear but are fully equivariant. EGNN out-performed tensor product methods despite not being fully equivariant. We extended EGNN to be fully equivariant and show that this further improves results. Thus, one wants to be fully equivariant (tensor-product or group-conv type) and be maximally non-linear (message passing type). We call such class of methods non-linear convolutions. We further observe, that in our steerable framework we can even further push performance by incorporating more geometric quantities into the the architectures, such as velocities and forces.

---

> ### Author Response · Authors · 2021-11-18
> **Response to Reviewer zF5H (1)**
>
> Thank you for the detailed review.
> We fully agree that some of the introduced concepts are hard to convey, and as discussed in the general response, we have therefore included pseudocode for SEGNNs and various ablations in the appendix. This hopefully answers a few questions and makes it easier to understand differences between various architectures.
>
> What follows are inline comments to your review.
>
> * `How to understand that using the spherical harmonic embedding of relative positions is including physical information? Is that a pure geometric information?`
> We differ between two kind of (vector-valued) quantities: geometric such as position or relative position, and physical such as force, spin, or velocity. The distinction comes more from physics, a position is normally referred to as geometric and a velocity as physical property. But for the algorithm it does not make a difference. Both quantities are vectors and can be encoded via spherical harmonics following formula (4) in the main paper. In the SEGNN algorithm (Algorithm 3) we see how these quantities can be used as edge attributes (a_ij) or node attributes (a_i). Relative positions or relative forces do not make sense as node attributes, but forces or velocities do. Our algorithm is the first where we can steer the node updates by leveraging these quantities.
>
> * `How to include relative force or relative momentum in the molecular tasks, e.g., the QM9 and OC20 experiments used in the paper.`
> This is a very good question since there are datasets (especially physics simulation) where this might be needed. Unfortunately such quantities are absent in the QM9 and OC20 datasets. They would probably help to boost SEGNN performances. Assumed that there were given one can encode these vector with spherical harmonics as it is done for the relative positions in Algorithm 3 and then add these two encodings as it is done for the node attributes.
>
> * `I assume that EGNN should be an important baseline of this paper, but seems that EGNN has not been reported on OC20.`
> That is a very good catch. In Table C.4 in the appendix SEGNN with la=0, lf=0 reduces to the special case of an EGNN. We renamed that properly. Please note that we can only do this comparisons on the validation dataset since submission to the official challenge are limited. But we fully agree, the EGNN baseline is very important. We have therefore also added SE_linear runs to Table C.4 in the appendix to further ablate our architecture design. As with all other experiments that we did, SEGNNs improve upon EGNNs.
>
> * `The performance of the proposed method is not very competitive. For example, UNiTE largely outperform baseline methods (SphereNet, PaiNN, etc) on QM9, but the performance gain of proposed SEGNN is marginal. Similarily, GemNet (Klicpera et al., NeurIPS 2021) and NoisyNode (Godwin et al.) perform well on OC20 IS2RE.`
> We fully agree with that statement. Not every model is suited for every dataset in the same way (we are happy to include UNiTE numbers upon publication). SEGNNs are designed for vector valued inputs and outputs, as well as available geometric and physical information. Only position information is available for the QM9 and the OC20 dataset. We would argue that despite the absence of these characteristics in most categories of QM9 and OC20 SEGNN perform on par or slightly worse than several state of the art models. Datasets for which SEGNNs are particularly suited are physical and fluid simulation datasets. An example is the n-body dataset where we have highlighted the benefits of SEGNNs. To stress this point more clearly, we have added another experiment to the paper draft with a 100 body gravitational dataset, highly relevant e.g. for astrophysics simulation. We want to stress once more that we did experiments especially on QM9 to have all these comparisons to existing methods even if it is not our number one choice dataset due to missing characteristics.
>
> * `Considering that the recent GemNet is also a powerful equivarient GNN, would the authors make a more comprehensive discussion to explain the pros/cons between these two methods?`
> We have added GemNet to the related work section. DimeNet(++) uses neighbour-neighbour information, whereas GemNet uses two-hop information even. While it is clearly beneficial that such higher-order neighborhood information is exploited (on the cost of longer runtimes) for e.g. most molecular datasets, we argue that if geometric and physical information is available it may be more efficient to directly utilize it in a simple first-order message passing scheme, as done in our method. We test this on the newly included gravitational dataset.

---

> > ### Comment · Reviewer_zF5H · 2021-11-19
> > **Thanks for the response**
> >
> > I appreciate the author's detailed clarification and introduction. Most of my concerns have been solved, thus I'd like to raise my score to 6. While I still have minor concern about the significance of performance gain of the proposed method, I agree with authors that not every model is suited for every dataset in the same way, and SEGNN would bring benefits in specific scenarios (could be demonstrated in the new added simulation experiment). Please kindly ensure that the nice explaination in the response would be included in the next version, report the mean MAE/EwT over subsets on OC20, and bold the best performance in C.4.
> >
> > A quick question: the GemNet-T / GemNet-dT with only one-hop information perform well too on OC20, therefore the benefits may not came from higher-order neighborhood information. Thus I still confuse with my question 6.

---

> > > ### Author Response · Authors · 2021-11-19
> > > **Response to quick question**
> > >
> > > Thank you for making us look into that direction. We are looking into a precise formulation to better establish the connection with Gemnet (and Dimenet) in a new paper draft, where we of course also include your other points (closed-off test set for OC20 C.4, bold figures C.4).
> > > This is a very interesting topic, and we want to elaborate on that a bit more.
> > >
> > > In the meantime, this is a possible explanation we have so far:
> > >
> > > Gemnet establishes a nice connection between steerable methods and directional message passing like Dimenet++ and Gemnet. The difference being that one can think of the steerable methods as assigning functions/feature maps on SO(3) at each node which has location $\mathbf{x}\in\mathbb{R}^d$. The graphs can then be seen as semi-sparse feature maps in $SE(3)=\mathbb{R}^3 \times SO(3)$, where sparsity is in the sparse locations $\mathbf{x}$, but at each location we have a dense signal on $SO(3)$. These signals on SO(3) are described by the steerable vectors (the Fourier trafo on SO(3) allows to map between steerable vectors -the Fourier coefficients- and functions on SO(3)). The paper is quite technical, so forgive us if we misunderstood it, but if we understood correctly, then in Gemnet the viewpoint is taken of representing sparse S^2 signals at each node, where only at certain directions $n_i$ a feature embedding is present. Thus one can think of Gemnet as indirectly processing a sparse signal on $\mathbb{R}^3 \times S^2$, where now sparsity is both in the spatial part as well as in the spherical part. The sparse directions in S^2 are defined by relative positions of neighboring atoms. The set of sparse directions is called an equivariant mesh. I read the derivation in Eq. 6 of the Gemnet paper as a regular group convolution, in which the integral is reduced via the sparsity of the signal, and the fact that due to processing of spherical data the kernel is constrained to be zonal.
> > >
> > > Thus, where our approach relates to the family of steerable (non-linear) group convolution methods, I see Gemnet as belonging to the family of regular (non-linear) group convolutions.
> > > Then, difference in performance can be explained via the classic discussion between steerable vs regular group convolutions in which one often sees that regular group convolutions outperform steerable ones since their processing is less restrictive. I.e. in regular group convs one can use point-wise non-linearities such as ReLU or Swish, whereas in the steerable case on is limited to gated non-linearities or other invariant activation functions. We believe that the success of Gemnet, partly lies in this line of reasoning, that through a clever construction they designed operators that are equivariant (akin to the regular group convolutions) without being constrained to use a particular class of activation functions. On top of this, their architectural design is more tuned towards the problem, e.g. Gemnet like Dimenet is not equivariant for vector valued inputs and outputs; whereas in our case we used a somewhat one-size-fits-all approach by using a similar simple architecture on a variety of tasks.

---

### Official Review · Reviewer_SAGC · 2021-11-08

**Correctness:** 4
**Technical Novelty And Significance:** 2
**Empirical Novelty And Significance:** 2
**Recommendation:** 6
**Confidence:** 4

**Main Review:**

The paper is well written. The main strengths of this paper are the strong empirical results on recently introduced benchmark (OC20) for learning on small molecules. Authors put down a good effort to jot connection between various convolutional and message passing algorithms in the main paper. Although I enjoyed reading the manuscript I think it would benefit from a clearer presentation of the novel technical contributions. For example, some form of comparison using pseudo code will help readers.

The overall contribution and novelty is minimal. As pointed earlier, SEGNNs is an extension of EGNNs (Satorras et al., 2021) wherein the linear layer and nonlinear activation within MLPs are replaced by their steerable version. The steerable linear transformations was earlier proposed in (Anderson, Fuchs, Thomas) while steerable activation is extended from (Wieler).

Other drawback and feedback:

1. For comparison on QM9, SEGNN performs better than EGNN on first six properties. However, SEGNNs performance on last six properties (non-energy ones) is much worse than previous state-of-the-art. Author points that more involved architecture might perform better. Given the level of complexity in SEGNN, I feel SEGNN is quite involved. How about increasing order or cutoff radius ?
2. From an ablation study on cutoff radius and order provided in Table C.2, I noticed that result improves considerably on increasing the cutoff radius. Can you also provide ablation results for higher cutoff radius (say at 7, 10, 15 A) ? Will lower order and increased cutoff radii give similar result ?
3. I suggest that for Table 2 and elsewhere, please present all baselines and results with fixed number of decimal points. This makes it easier to compare across models. Also, for each of the properties, please highlight the best model in bold.
4. The model claims scalability to large molecules due to the use of fewer MP steps. But the application of same on large molecule datasets is missing. OC20 contains maximum of 70 atoms which is not large enough. Some datasets such as GEOM-Drugs contains max. of 180 atoms. Moreover, given that the model is applicable to any physical system, I would like to see the performance on fluid simulation (A).
5. What is the actual difference between SE(non-linear) and SEGNN_G ?
6. The results on N-body system in Table 1 suggest that in comparison to EGNN, SEGNN (order 1) significantly reduce error but at the cost of ~10x slower runtime.
7. What is odd parity ?

A. Learning to Simulate Complex Physics with Graph Networks, ICML 2020.

**Summary Of The Paper:**

This work extends the concept of steerable kernel to nonlinear transformations in MLP. The steerable MLPs are then incorporated into recently introduced equivariant message passing layers thus forming Steerable E(3) Equivariant GNNs (SEGNNs). At the outset SEGNNs are generalisation of EGNNs (Satorras et al., 2021) with non-isotropic message functions utilising relative pose information instead of mere distances. When applied to molecular property prediction tasks, it achieves mixed results on QM9 data and state-of-the-art results on IS2RE task of OC20 dataset.

**Summary Of The Review:**

Drawing connection with steerable network, this work attempts to rectify a missing link in prior art for equivariant network. Although, the technical contribution is minimal, the paper would have stand out if the empirical results were strong. I suggest that authors improve upon their results and further demonstrate wide applicability.

---

> ### Author Response · Authors · 2021-11-18
> **Response to Reviewer SAGC**
>
> Thank you for the detailed review.
> We think that the suggestion of pseudocode is a great idea.Therefore, as indicated in the general response to make the novel ideas more concrete we added pseudocode for the SEGNN architecture, as well as for ablated architectures to the paper. The pseudocode should make the different ablations (linear vs non-linear, new activation function / new node update) easier to understand. In the SEGNN pseudocode, it becomes clear that only via steerable node updates such information can be included.
> Regarding scalability and broader applicability to physical systems, we did several experiments on a new physics based simulation experiment (see general response). We created a new gravitational simulation dataset which consists of 100 particles, interacting via gravitational force. We include this new dataset, as well as comparisons to baselines in the new paper draft. We see such physical simulations as very likely and strong future application of our method. A direct application to much larger physical simulations such as Sanchez-Gonzalez et al. is however a bit beyond the scope of this paper.
> What follows are inline comments to your review.
>
> * `For comparison on QM9, SEGNN performs better than EGNN on first six properties. However, SEGNNs performance on last six properties (non-energy ones) is much worse than previous state-of-the-art. Author points that more involved architecture might perform better. Given the level of complexity in SEGNN, I feel SEGNN is quite involved. How about increasing order or cutoff radius.`
> We consider e.g. attention mechanism (as used in EGNNs) or neighbour-neighbour interactions (as used in DimeNet and SphereNet) as more involved with respect to architecture complexity. Atom number embedding and relative position information might not be enough for complex tasks like energy targets, we try to discuss this very openly. Please keep in mind that SEGNNs are designed for vector valued inputs and outputs, as well as available geometric and physical information. The less of such is available the more other information might be necessary (i.e. neighbour-neighbour information). We added ablation with larger cutoff radius and l=1 to the table, thanks for pointing this out. This is an important comparison.
>
> * `From an ablation study on cutoff radius and order provided in Table C.2, I noticed that result improves considerably on increasing the cutoff radius. Can you also provide ablation results for higher cutoff radius (say at 7, 10, 15 A) ? Will lower order and increased cutoff radii give similar result ?`
> We added ablation with larger cutoff radius and l=1 to the table. Higher order cutoff radius and l>=1 gets computationally intractable on the OC20 dataset. Some preliminary test runs however showed that results are not improving by further increasing the cutoff (as also reported by other methods). We tried to do the cutoff ablation very thoroughly on QM9. We extended the cut-off radius experiments with different steerability orders for OC20 to radius = 3A, 5A and for QM9 to radius = infty, 2.5A, 5A.
>
> * `I suggest that for Table 2 and elsewhere, please present all baselines and results with fixed number of decimal points. This makes it easier to compare across models. Also, for each of the properties, please highlight the best model in bold.`
> Thanks for pointing this out. Done.
>
> * `The results on N-body system in Table 1 suggest that in comparison to EGNN, SEGNN (order 1) significantly reduce error but at the cost of ~10x slower runtime.`
> Yes this is true and a general downside of steerable methods. The reason is the much higher computation complexity of the Clebsch-Gordan tensor product. For l>0 more operations are needed for one weight. This can be seen in Eq.(5) of our paper.
> We have added a small section to the appendix A.5 and discuss how higher orders of the Clebsch-Gordan tensor product effects computational complexity and therefore runtimes. On the other hand, steerable methods allow for much more data efficient training. We have added Table C.2 in the appendix to demonstrate that. Note that e.g. with a third of the training set size SEGNNs obtain better results than the EGNNs.
>
> * `What is the actual difference between SE(non-linear) and SEGNN_G?`
> Very important question. Hopefully, the difference is more clear now with the introduction of the pseudocodes in the Appendix (Algorithm 2 (SE_non-linear) and Algorithm 3). The difference is in the node update networks. Additional information (SEGNN_G+P) can then be included by adding more node attributes.
>
> * `What is odd parity ?`
> Parity is the property of how things change under reflection. Even parity means that quantities do not change under reflection (as it is for scalar quantities), odd parity means that quantities change (as vectors does when mirrored).

---

> ### Comment · Reviewer_SAGC · 2021-12-09
> **Bumping my score**
>
> Sorry for my very very late response. I appreciate authors detailed explanations and their effort on improving readability. This work stands out more as a complete guide for "what, how" on equivariant message passing layers. This should act as a catalyst for future works and hence the increase in my score.

---

### Author Response · Authors · 2021-11-18
**General response**

We thank all the reviewers for the constructive feedback. We were pleased to see that a number of reviewers found the paper thorough and well written, the work novel and reasonable, the exposition very clear throughout, the result section strong, the formulation general, the ablations thorough, and the added code clean.

We greatly appreciate the comments and suggestions by the reviewers as we believe their incorporation led to valuable improvements to the paper in terms of clarity and they further improved the thoroughness of the experiments. The following lists the **main improvements to our paper in response to the reviewers queries**:

1) We extended the n-body experiments to test the data efficiency of SEGNN on different number of training samples (Table C.2 in appendix)
2) We added pseudocode for SEGNN architectures, for linear, and for non-linear ablated architectures in a new subsection in the appendix (C1)
3) We designed a completely new experiment on the simulation of many body (100 bodies) gravitational interactions, proposing a new dataset. We see this as an additional important showcase to test SEGNNs for their applicability to physics (simulation) tasks. There is quite a difference between 5-body and 100-body interactions, we want to demonstrate the scaling properties of SEGNNs.
4) We ran SEGNN, as well as MPNN and EGNN ablation on the new gravitational N-body dataset. We tried to implement Dimenet as well, but faced instabilities during training which we intend to resolve for the camera ready version, but cannot report yet. We currently report the experiments and implementation in Appendix C3.
5) We did the requested ablations and improvements of the presentation of QM9 and OC20 results.
6) We added a short section about gated nonlinearities in App C.1 and marked clearer in the main paper that we only use gated nonlinearities so far.
7) We added a short section in the App. A.5 where we discuss how higher orders of the Clebsch-Gordan tensor product affect computational complexity and therefore runtimes.
8) We have improved Sec 3 (presentation of non-linear convolutions and related work) with a new introduction and small modifications (and references to the details in the appendix) to improve readability.

We address all reviewers individually, but here already address two points which are commonly shared among reviewers.

**Presentation of novelty of the paper, linear vs non-linear convolution, new class of activation functions**:
There is an abundance of SE(3) equivariant point conv and graph networks in literature. We rigorously analyzed many of these methods and put them into a unified framing. We thereby identify steerable non-linear messages and steerable node updates as most important components. The latter makes it possible to inject geometric and physical quantities into the node updates, and therefore can be seen as a new type of steerable activation function. In short, we are the first paper using CG tensor products as non-linear mapping between steerable feature spaces for both message and node updates, where the non-linear operation can be arbitrarily equipped with geometric and physical quantities. Non-linear steerable updates conditioned on geometric quantities is unique and has not been done before. It proves a very powerful tool.
To make these ideas more concrete we added to the paper pseudocode for the SEGNN architecture as well as for other ablated architectures. The pseudocode should make the different ablations (linear vs non-linear, new activation function / new node update) more clear. SEGNNs might look like a straightforward extension of EGNNs, but are tailored towards complex physics (simulation) datasets where vector valued information is available which needs to be carefully injected into equivariant architectures.
Concluding, we want to highlight that we explicitly strive to present our method in connection to as many methods as possible without obfuscation or additional complexity. Most notably, it is not possible in any of the related methods to incorporate geometric and physical quantities. Our modifications can make a big difference when processing physical (simulation) datasets.

**Experiments, applicability to any physical systems**.
SEGNNs are designed especially for datasets with vector-valued input and output information and available geometric and physical quantities. Although most of this information is absent in the QM9 and OC20 dataset (except position information), SEGNNs perform mostly on par with state of the art methods.
We see it as our scientific duty to compare SEGNN performance also on QM9, especially since many, many methods are benchmarked there.
To further showcase the potential of SEGNNs, we created a new additional gravitational simulation dataset which consists of 100 particles. Together with the release of our code base, we consider this as an important step-up towards broader application of SEGNN in physics (simulation) tasks.

---

### Decision · Program_Chairs · 2022-01-20

**Decision:**

Accept (Spotlight)

**Comment:**

This work combines steerable MLPs with equivariant message passing layers to form Steerable E(3) Equivariant GNNs (SEGNNs). It extends previous work such as Schnet and EGNNs, by allowing equivariant tensor messages (in contrast to scalar or vector messages). The paper also provides a unifying view of related work which is a nice overview for the ML community. It is overall well written, but would benefit from further revision to improve readability in some parts (in particular section 3, cf. reviews).
It shows strong empirical results on the IS2RE task of the OC20 dataset and mixed results on the QM9 dataset.